# The Carbon Emission Characteristics and Reduction Potential in Developing Areas: Case Study from Anhui Province, China

**DOI:** 10.3390/ijerph192416424

**Published:** 2022-12-07

**Authors:** Kerong Zhang, Liangyu Jiang, Yanzhi Jin, Wuyi Liu

**Affiliations:** 1School of Business, Fuyang Normal University, Fuyang 236037, China; 2School of Biological Science and Food Engineering, Fuyang Normal University, Fuyang 236037, China

**Keywords:** carbon emission, emission reduction, carbon emission inventory, environmental Kuznets curve, STIRPAT model, ridge regression, scenario analysis, scenario prediction

## Abstract

Global warming and world-wide climate change caused by increasing carbon emissions have attracted a widespread public attention, while anthropogenic activities account for most of these problems generated in the social economy. In order to comprehensively measure the levels of carbon emissions and carbon sinks in Anhui Province, the study adopted some specific carbon accounting methods to analyze and explore datasets from the following suggested five carbon emission sources of energy consumption, food consumption, cultivated land, ruminants and waste, and three carbon sink sources of forest, grassland and crops to compile the carbon emission inventory in Anhui Province. Based on the compiled carbon emission inventory, carbon emissions and carbon sink capacity were calculated from 2000 to 2019 in Anhui Province, China. Combined with ridge regression and scenario analysis, the STIRPAT model was used to evaluate and predict the regional carbon emission from 2020 to 2040 to explore the provincial low-carbon development pathways, and carbon emissions of various industrial sectors were systematically compared and analyzed. Results showed that carbon emissions increased rapidly from 2000 to 2019 and regional energy consumption was the primary source of carbon emissions in Anhui Province. There were significant differences found in the increasing carbon emissions among various industries. The consumption proportion of coal in the provincial energy consumption continued to decline, while the consumption of oil and electricity proceeded to increase. Furthermore, there were significant differences among different urban and rural energy structures, and the carbon emissions from waste incineration were increasing. Additionally, there is an inverted “U”-shape curve of correlation between carbon emission and economic development in line with the environmental Kuznets curve, whereas it indicated a “positive U”-shaped curve of correlation between carbon emission and urbanization rate. The local government should strengthen environmental governance, actively promote industrial transformation, and increase the proportion of clean energy in the energy production and consumption structures in Anhui Province. These also suggested a great potential of emission reduction with carbon sink in Anhui Province.

## 1. Introduction

Climate change is any change in the temperature or water levels that affects the Earth for a long period of time. Global climate changes, such as high temperature, heat waves, rising sea levels, and other extreme weather caused by massive carbon emissions, are becoming increasingly prominent [1,2,3,4]. Humans today are experiencing rapid and unexpected climate changes in temperature whose impacts of climate change do not stop at hotter temperatures. Undoubtedly, it is imminent to promote both developed and developing countries to work together to reduce global carbon emissions for the terrible greenhouse effects [1,3,4,5]. In developed countries, statistics revealed that the emissions of CO2 and other greenhouse gas (GHG) generated from energy use in economic activities have largely decreased in recent years [6,7,8]. For example, it is shown that carbon dioxide (CO2) is much produced and sharply reduced in the United States and United Kingdom since the year 2007 [6,9,10], whereas CO2 emissions from energy use clearly decreased in the European Union (EU) in 2020 [8]. In 2020, there was a great decrease in the CO2 and other greenhouse gas emissions in EU since 2008 [7]. The economy greenhouse gas emissions, compared with 2019, there was a 9% decrease, and compared with 2008, there was a 24% decrease [7]. Recently, to tackle climate change, the European Parliament adopted the European Climate Law, which raises the EU’s 2030 emissions reduction target to at least 55% from 40% and makes climate neutrality by 2050 legally binding [11]. At present, climate change is already affecting the world, with extreme weather conditions such as the drought, heat waves, heavy rain, floods and landslides becoming more frequent, including in Europe. Other consequences of the rapidly changing climate include rising sea levels, ocean acidification and loss of biodiversity [8,11,12,13]. In order to limit global warming to 1.5 degrees Celsius—a safe threshold suggested by the Intergovernmental Panel for Climate Change (IPCC), carbon neutrality by mid-21st century is essential [12]. This target is also laid down in the Paris agreement signed by 195 countries, including the EU [12]. With the European Climate Law, the EU has committed to carbon neutrality by 2050 in December 2019 [11,12,13]. This target will be reached through the European Climate Law that sets climate neutrality into binding EU legislation [12]. As a developing country, China is regarded as one of the world’s largest carbon emitters with much consumption of fossil energy in the world. Recently, China pledged to cut CO2 emissions and boost global confidence in climate change; it needs to overcome greater challenges than developed countries to meet its carbon-neutral target. The Chinese government has implemented corresponding strategies and measures soon to reduce carbon emissions with a high attention. On September 21, 2021, Chinese President Xi Jinping advised at the 76th United Nations General Assembly to achieve the carbon peaking by 2030 and its carbon neutrality by 2060 [13,14,15]. Under the overall requirements of Chinese goals of dual carbon and carbon neutrality, understanding the overall carbon emission level and carbon sink potential is the national basis and core element of promoting low-carbon development in the China [16,17,18,19,20,21,22,23,24,25,26,27,28,29,30,31]. Similarly, under the regional “14th Five-Year Plan”, the provincial government is devoted to reduce its carbon emission to 65% of that in 2005 by 2030 in Anhui Province. In particular, some metropolises in Anhui Province, i.e., Hefei, Huaibei, Huangshan, Lu’an and Xuancheng, are selected as the third batch of the national low-carbon pilot developing cities. As one of the largest providers of steel, automobiles, electricity, coal, and agriculture products in China, this has greatly promoted the present low-carbon development process in Anhui Province. On this basis, the carbon emission inventory in Anhui Province was systematically compiled in this study. Thus, the study analyzed and explored the overall carbon emission level and carbon sink potential in Anhui Province. It might also clarify the important sources of carbon emissions and identify the crucial areas and main points of carbon emission reduction. Subsequently, the overall carbon emissions and regional carbon sink capacity of Anhui Province were calculated from 2000 to 2019. Meanwhile, carbon emission changes and carbon emissions of various industrial sectors were compared and discussed too.

Agriculture, as the primary industry, is the second largest source and driver of carbon emissions in the world, and its carbon emissions in agronomy and animal husbandry cannot be ignored [16,25,26,27,28,29,31,32,33,34,35,36,37,38,39,40,41,42]. In calculating the level of agricultural carbon emissions in Indonesia, scholars had proved that there were an inverted “U”-shaped assumption of the environmental Kuznets curve, carrying out rice cultivation and increasing the ratio of renewable energy use could reduce the impact of carbon emission [32].When Leitão and Balogh (2020) measured the relationship between Portuguese agriculture and energy consumption, they found that energy consumption played a role in promoting agricultural carbon emissions [33]. The overall difference of China’s agricultural carbon emission showed a “U”-shaped correlation with economic development, and the carbon emission of each province showed a downward trend. However, the downward trend was different in the differentiated provincial regions, showing a strong spatial agglomeration distribution pattern [34,35,36]. At the same time, China’s agricultural carbon emissions will continue to increase in the future. The indicator GDP per capita is regarded as the main driving force for its growth. Increasing agricultural input will help reduce carbon emissions. This requires the Chinese government to formulate effective agricultural development policies to ensure the high-quality agricultural development [36]. In practice, environmental changes caused by carbon emissions are not conducive to the development of agriculture [37], whereas environmental pollution caused by agricultural carbon emissions is even the main cause of health problems in relatively poor areas in China [38], which requires us to take effective measures to reduce agricultural carbon emissions. In view of the gradient decreasing pattern of “west > central > east > northeast” of China’s agricultural carbon emissions and the gradual narrowing of regional differences, it is particularly necessary to adopt some differentiated emission reduction policies 40]. For example, we might plan to speed up the adjustment of agricultural structure to promote the progress of agricultural technology [26]. Simultaneously, we might try to explore curtain pathways to promote the agricultural carbon-peaking by means of technology and finance [22,40], or to promote the clean agricultural production and expand the opening of agriculture to the outside world [41,42].

It is well known that industry (i.e., the secondary industry) is the global largest source and driver of carbon emissions [43,44,45]. Industry is also one of the main sources of carbon emissions. Excessive CO2 caused by industrial energy consumption has become the crucial factor in environmental degradation and restricting the sustainable development of industry. Increase in the usage of low-carbon energy and decrease in the usage of high-carbon energy might be changed by changes in the industrial structure [46]. In recent years, carbon emissions from transportation are growing rapidly. With the rapid development of China’s aviation market, aircraft round-trip fuel has pushed its carbon emissions to a higher level. This requires the government to strengthen the guidance of the tertiary industry, especially the transportation industry, and promote its green and low-carbon development [47]. With the rapid development of the economy, the process of urbanization has been accelerated in the world, which has led to not only the gradual increase in the annual energy use of cities, but also serious energy shortages and environmental pollution [48]. Industrial cities and the energy supply areas accounted for 80% of global energy consumption and 60% of carbon emissions. From the perspective of urban stakeholders, they could better promote the development of a sustainable low-carbon society [49]. Due to the increase in income and population, people’s living standards have been greatly improved. Ismail et al. (2020) took whole Southeast Asia as the research object and found that the changing relationship between its economic development and energy consumption was in line with the inverted “U” hypothesis of the environmental Kuznets curve [50]. Furthermore, they suggested that formulating appropriate policies could ensure energy balance and improved the local environment quality [50]. Another study on the Beijing-Tianjin-Hebei region analyzed the direct and indirect carbon emissions caused by urban and rural residents’ consumption [51]. It was found that residents’ consumption level and carbon emission were the main driving factors [51]. Urban employment, rapid urbanization and the two-child policy also contributed to the generation of carbon emissions [51,52]. However, by adjusting the consumption structure of residents and promoting technological progress in energy-intensive industries, the growth rate of indirect carbon emissions from household consumption might be slowed down [52]. Rural and mountainous areas are more dependent on forests for daily fuel, fodder and food, whereas household consumption carbon emissions could be reduced by improving their utilization of biomass and developing cost-effective solar water heating systems [53]. With the rapid growth of per capita GDP, China’s food consumption has increased significantly, and the resulting carbon emissions have also increased rapidly. In terms of the dietary structure, China is recently shifting towards a more animal-based Westernized diet, which also affects the household carbon emissions [54]. In order to reduce the carbon emissions from household food consumption, Japan was analyzed and it was found that more than 60% of the food carbon footprint occurred in the food production process, in which the wholesale and retail carbon emissions accounted for as high as 38% [55]. By optimizing the supply chain managing and encouraging the consumption of local fruit, vegetables and meat could reduce carbon emissions from household food consumption. According to the characteristics of the local diet, the government promotes its eating habits to develop in a more sustainable direction, which can also reduce the carbon emissions caused by household food consumption [56]. Greenhouse gases such as CH4 and CO2 caused by ruminants have made a significant contribution to global warming, and reducing the production of ruminant greenhouse gases is crucial to mitigate global warming [57]. Among them, CH4 was released into the atmosphere by the intestinal peristalsis of ruminants accounted for 30% of the total atmospheric methane and it was one of the main sources of carbon emissions [57]. Carbon emissions from intestinal peristalsis might be reduced by improving feeds and other methods [58]. With the development of urbanization and the improvement of people’s living standards, a large amount of food waste and decoration waste are continuously produced, and the carbon emissions generated by them are increasing rapidly [59]. Thiel et al. (2021) proposed to reduce the carbon emissions caused by food waste through some feasible solutions such as recycling and moderate composting in a large hospital in New York City [60]. Olujobi et al. (2022) suggested to improve the Nigerian power sector while maintaining environmental protection by converting waste to energy [61]. Carbon sink is the process, activity, or mechanism of absorbing carbon dioxide in the atmosphere through afforestation, vegetation restoration and other measures, so as to reduce concentration of greenhouse gas in the atmosphere [62,63,64,65]. It is an important part of the ecosystems and an important way to achieve the regional carbon neutrality [66,67,68,69,70,71,72,73,74,75,76]. In the long-term grain crop planting process, grain crop production and forest coverage had a significant effect on carbon emission absorption [64,65]. In fact, this case is of enormous significance to Anhui Province, one of the largest producing areas of grains and other agriculture products in China.

When reviewing and summarizing the relevant studies on carbon emissions, it was found that energy consumption, food consumption, ruminants and waste disposal of the three major industrial sectors and residents’ lives are the main sources of carbon emissions, while forests, grasslands and crops are important components of the carbon sink [62,63,64,65,71]. Starting from these aspects, we can comprehensively measure the status quo of regional carbon emissions and grasp the law of changes. Based on this, we draw on the compilation method of carbon emission inventory, mainly referring to the 2006 version of IPCC (Intergovernmental Panel on Climate Change) Guidelines for National Greenhouse Gas Inventories (IPCC website available online: https://www.ipcc-nggip.iges.or.jp/public/2006gl/index.html, accessed on 3 January 2022) released on 23 October 2006. In the IPCC guidelines, carbon emissions mainly come from the following five aspects: energy activities, industrial production, agricultural activities, land use change and forestry, and waste disposal, which are in line with the research directions of many of the above mentioned scholars. This study also refers to the research conducted by related scholars on carbon emission inventories. For example, the current status of China’s carbon emissions was calculated and explored from the city level based on a high-resolution carbon emission database [72]. The greenhouse gas accounting method of energy consumption was employed to measure the carbon emission level of Xiaolan town in Zhongshan, China [73]. The carbon emission inventory of 11 cities was compiled in the greater bay area and their surrounding areas based on the IPCC regional carbon emission accounting method [74]. At the same time, the IPCC regional carbon emission accounting method was adopted to calculate the carbon emission of 15 cities in Sichuan Province, China [75]. The carbon emission level of Wroclaw (Poland) was measured and analyzed by compiling the carbon gas emission inventory at the city level [76]. These researchers found the low-carbon development pathways by evaluating the changing laws of carbon emissions on the basis of compiling carbon emission inventories.

There are still some shortcomings in the existing research. Relevant scholars only account for major energy sectors when compiling carbon emission inventories, which makes them unable to reflect the overall carbon emission level. It is convenient in research by only considering the source and size of carbon emissions and ignoring the regional carbon sink capacity. However, it cannot obtain a more comprehensive understanding of the carbon balance development status. Furthermore, there are presently few systematical research reports on the carbon emissions in developing areas such as Anhui Province, China. In order to comprehensively measure the levels of carbon emissions and carbon sinks in Anhui Province, this study adopted the IPCC (Intergovernmental Panel on Climate Change) carbon accounting method and other specific methods to analyze and explore the datasets from the following suggested five carbon emission sources of energy consumption, food consumption, cultivated land, ruminants and waste, and three carbon sink sources of forest, grassland and crops to compile the carbon emission inventory in Anhui Province. It was found that there were significant differences in the growth of carbon emissions among various industries. The ranking of measured data was secondary industry (Industry) > tertiary industry (Services) > primary industry (Agriculture) in terms of carbon emissions, while the measured data were ranked as tertiary industry (Services) > secondary industry (Industry) > primary industry (Agriculture) in terms of carbon emission growth rate. On this basis, regional differences of carbon emissions were analyzed and evaluated, while the reason for the differences was explored in the carbon emission levels of various provincial departments. The STIRPAT (stochastic impacts by regression on population, affluence, and technology) model was used to analyze the relevant factors affecting carbon emissions, and the future carbon emission was predicted based on the scenarios to judge whether it can achieve the goals set by the state. Finally, the study supplemented and probed into the following questions with the refined low-carbon countermeasures proposed for Anhui Province: Was there an environmental Kuznets curve in Anhui Province? How about the impact of urbanization on the change of carbon emission in Anhui Province?

## 2. Materials and Methods

### 2.1. Study Area, Carbon Emission Inventory, and Data Retrieval

#### 2.1.1. Study Area

The study area is the whole region of Anhui Province, located on mainland, China. It is close to the river and the sea. It has the Yangtze River waterway inside and the economic radiation of coastal areas outside. Anhui Province is situated in the transitional area between warm temperate zone and subtropical zone, with huge social economic development potentials. It also crosses many large rivers, such as the Huaihe River, the Yangtze River, and the Xin’an River. There are currently 16 provincial cities, 9 county-level cities, 50 counties, and 45 municipal districts in Anhui Province. The terrain of Anhui Province is comprised of plains, hills and mountains. Nevertheless, north of Anhui is part of the area of warm temperate semi humid monsoon climate, while the south of Anhui belongs to the area of subtropical humid monsoon climate.

#### 2.1.2. Carbon Emission Inventory and Data Retrieval

In addition to carbon emissions caused by human activities, there are also different types of carbon emissions in nature itself, as well as carbon sink capacity, which together constitute the carbon balance of the earth’s ecosystems. In the “2006 IPCC National Greenhouse Gas Inventory Guidelines” adopted by IPCC, carbon emission sources are divided into the following five aspects: energy activities, industrial production, agricultural activities, land use change, forestry and waste disposal.

First, this study divided the carbon emission sources into five aspects, i.e., energy consumption carbon emission, resident food consumption carbon emission, cultivated land carbon emission, ruminant animal carbon emission and waste carbon emission. Among the indexes of carbon emissions, those of energy consumption were divided into production and consumption carbon emissions and living consumption carbon emissions, which was mainly employed to calculate the carbon emissions caused by fossil energy consumption and electricity energy consumption in the daily production and life. Simultaneously, those indexes of carbon sinks were divided into the following three parts, i.e., forest, grassland and crops.

Next, the study adopted the IPCC carbon accounting method from the above defined carbon emission sources (i.e., energy consumption, food consumption, cultivated land, ruminants and waste, and three carbon sink sources of forest, grassland and crops) to compile the carbon emission inventory in Anhui Province.

The relevant data were mainly retrieved from the official released 2001–2020 versions of the *China Energy Statistical Yearbook* and the *Anhui Provincial Statistical Yearbook* in the oversea website of China National Knowledge Infrastructure (CNKI, http://tongji.oversea.cnki.net/oversea/engnavi/navidefault.aspx, accessed on 3 January 2022). The updated CO2 emission inventory can be found in the website of China Emission Accounts and Datasets for free download (https://www.ceads.net/, accessed on 3 January 2022). The research time was set at 2000–2019, whereas the relevant datasets could only be retrieved from the 2001–2020 versions of the *China Energy Statistical Yearbook* and the *Anhui Provincial Statistical Yearbook*. Herein, some relevant parameters were referred to the previous reports [72,73,74,75,76,77,78].

### 2.2. Calculations of Carbon Emissions

#### 2.2.1. Carbon Emissions from Energy Consumption

In Equation (1), *C_ee_* is the total amount of carbon emissions from energy consumption, while *Q_ei_* denotes the consumption of the *i*-th type of energy.
(1)$$C_{ee}=\sum_{i=1}^{m}\left(Q_{ei}\times u_{i}\times \alpha_{i}\right)$$

Simultaneously, *u_i_* is the coal conversion coefficient of the *i*-th type of energy, and *α_i_* means the carbon emission coefficient of the *i*-th category energy unit of standard coal. Fossil energy is mainly divided into coal, oil and natural gas, and its carbon emission coefficient refers to the reported “Guidelines for Compilation of Provincial Greenhouse Gas Inventories”. When calculating carbon emissions caused by the electricity consumption, we refer to the calculation method of secondary energy carbon emissions used by previous researchers [78]. The specific calculation process is shown in the following Formulas (2)–(4).
(2)$$EEV_{m}=\frac{\sum_{i}\left(FC_{i,m}\times u_{i}\right)}{EG_{m}}$$
(3)$$K_{m}=\frac{EEV_{m}-ECV}{EEV_{m}}$$
(4)$$EF_{m}=\frac{\sum_{i}\left(FC_{i,m}\times EF_{co_{2},i}\right)}{EC_{m}}$$

In the expression, *EEV_m_* shows the energy equivalent value of electricity in the m-th year (kgce/kW·h, i.e., total energy consumption of standard coal in kilogram per kilowatt hour), and *ECV* means the energy calorific value of electricity (kgce/kW·h). *FC_i, m_* denotes the consumption of fuel *i* in electricity production in the mth year (kg or m^3^), *u_i_* markers the coal conversion coefficient of fuel *i* (kgce/kg or kgce/m^3^, i.e., total energy consumption of standard coal in kilogram per kilowatt hour or per cubic meter), and *EG_m_* means the provincial regional power production in the mth year (10^8^ kW·h). *EF_m_* shows the carbon emission factor of electricity in the mth year (tco_2_/10^4^ kW·h, the marker tco_2_ means the total carbon dioxide here), $EF_{co_{2},i}$ represents the carbon emission factor of fuel *i* (kgco_2_/kg), and *EC_m_* is the electricity consumption of the provincial regional consumer in the mth year (10^8^ kW·h). *K_m_* indicates the ratio of the provincial regional power production in the mth year, whereas *i* denotes the type of fossil fuel consumed in the electricity production of the provincial region in year m. The energy equivalent value of electricity was now published in the “China Energy Statistical Yearbook”, and the specific value used was 0.1229 (kgce/kW·h).

#### 2.2.2. Carbon Emissions from Residents’ Food Consumption

Carbon emission sources from food consumption of residents are different from those carbon emissions from energy consumption. The carbon emissions of residents’ food consumption consist of the following two parts, i.e., the carbon emissions of food consumption of the urban residents and the rural residents. The specific calculation formula is shown in Equation (5).
(5)$$C_{ed}=\sum_{i=1}^{n}\left(Q_{di}\times h_{i}\right)$$

Among them, *C_ed_* is the carbon emission from food consumption, *Q_di_* means the consumption of food i, and *h_i_* denotes the carbon emission coefficient of food i. Residents’ food consumption is mainly divided into grain, vegetables, drinks, fruit, poultry meat, aquatic products, vegetable oil, eggs, sugar and milk. The specific carbon emission coefficient is shown in Table 1.

#### 2.2.3. Carbon Emissions in Cultivated Land

As an important original source of carbon emissions is the agrarian production in agricultural areas, the methane is produced by anaerobic actions of the methanogens in human cultivated land activities, whereas carbon emissions come from the methane in cultivated land. Cultivated land is divided into paddy fields for the cultivation of rice and rape and dry land for the cultivation of wheat and other crops. Here, the carbon emissions naturally caused by soil in the process of cultivated land are concerned and calculated, while the carbon emissions caused by use of agricultural machinery are not involved. The calculation formula of carbon emissions is shown in the Equation (6).
(6)$$C_{ep}=(Q_{1p}\times C_{1p}+Q_{2p}\times C_{2p})\times 12/16$$

Among them, *C_ep_* shows the total carbon emission of cultivated land, and *Q_1p_* means the paddy field area. Meanwhile, *Q_2p_* and *C_2p_* are the dryland area and dryland methane emission coefficient, and *c_1p_* denotes the methane emission coefficient per unit area of paddy field, respectively, and 12/16 is the methane carbon emission coefficient (0.3126 tons/ha for paddy fields, 0.0069 tons/ha for dry land).

#### 2.2.4. Carbon Emissions from Ruminants

During the ingestion of food in ruminants, methane is produced by anaerobic fermentation of food in the stomach, which is excreted into the air with feces and intestines. The specific carbon emission calculation formula is shown in Equation (7):(7)$$C_{eb}=\sum_{i=1}^{k}\left(Q_{bi}\times g_{i}\right)\times 12/16$$

Here, *C_eb_* expresses the carbon emission of ruminants, *Q_bi_* is the number of *i*-th category animals, *g_i_* denotes the intestinal peristalsis and fecal methane emission coefficient of *i*-th category animals, and marker 12/16 indicates the methane carbon emission coefficient. The specific carbon emission coefficients are shown in Table 2.

#### 2.2.5. Waste Carbon Emissions

The carbon emission of waste mainly comes from wastewater and solid waste. The discharge of wastewater mainly produces methane, the solid waste treated by incineration mainly produces carbon dioxide, and the solid waste treated by landfill mainly produces methane. The specific calculation formula is shown in Equations (8) and (9):(8)$$C_{ew}=Q_{cod}\times 0.25\times 12/16$$

Here, *C_ew_* denotes the total carbon emission of wastewater, and *Q_cod_* is the quantity of chemical oxygen demand of wastewater. Specifically, 0.25 indicates the methane emission factor of unit wastewater, and 12/16 is the methane carbon emission coefficient.
(9)$$C_{er}=Q_{r1}\times 0.99945\times 0.45+Q_{r2}\times 0.167\times 0.285$$

Herein, *C_er_* shows the total amount of waste carbon emissions, and *Q_r1_* is the waste incineration amount. Particularly, 0.99945 indicates the waste carbon emission factor and 0.45 is the waste carbon content, while *Q_r2_* denotes the landfill amount, 0.167 is the waste methane emission factor, and 0.285 means the waste water content. Due to the difficulty in collecting relevant data, solid waste treatment only involves incineration and landfilling of domestic waste.

### 2.3. Calculations of Carbon Sinks

As an important way to maintain the carbon balance of ecosystems in nature, carbon sink capacity plays a crucial role in measuring the carbon emission reduction potential of a country or area achieving the regional sequestration and carbon neutrality. Considering the availability of indicators, the carbon sink capacity of Anhui Province is calculated from the fowling three aspects, i.e., forest, grassland and crops.

#### 2.3.1. Forest and Grassland Carbon Sinks

Forests and grasslands play an important role in natural carbon sinks and they are of great significance to maintaining the carbon balance of ecosystems. Its specific carbon sink formula is shown in Equation (10).
(10)$$C_{sf}=\sum_{i=1}^{2}K_{i}\times T_{i}$$

Here, *C_sf_* shows the carbon absorption amount of forest and grassland, *K_i_* is the area of forest and grassland, and *T_i_* means the carbon absorption rate of forest and grassland. In general, the carbon uptake rate of grassland is 0.948229 tons/ha, and the carbon uptake rate of forest is 3.809592 tons/ha in the released versions of the *China Energy Statistical Yearbook*. Since the grassland area data in Anhui Province are partially missing, but the overall change is not large. Thus, a few missing values were replaced by the average data of the adjacent years in the study.

#### 2.3.2. Crop Carbon Sink

The carbon sink of crops is mainly reflected in the absorption of carbon dioxide in nature through photosynthesis, which is converted into the energy and oxygen that they need. The specific calculation formula is shown in Equation (11).
(11)$$C_{sc}=\sum_{i=1}^{v}\left(S_{ci}\times Y_{ci}\right)=\sum_{i=1}^{v}\left(S_{ci}\times Q_{ci}/\omega_{i}\right)$$

Here, *C_sc_* shows the carbon sequestration of crops, while *S_ci_* is the carbon absorption coefficient of class i crops. Concurrently, *Y_ci_* indicates the output of class i crops, *Q_ci_* is the economic output of class i crops, and $\omega_{i}$ means the economic coefficient of class i crops. The specific carbon emission coefficient is shown in Table 3.

### 2.4. Prediction of Carbon Emission Reduction Potential

In order to further explore the relevant influencing factors of carbon emission in Anhui Province, the STIRPAT model was used to analyze the urbanization rate, industrial structure, energy intensity and GDP per capita that affect carbon emission. The influence degree of each factor on the carbon emission of Anhui Province was explored, and the scenario analysis method was used to further analyze the carbon emission reduction potential of Anhui Province under different scenarios, and explore the low-carbon development pathway of Anhui Province.

#### 2.4.1. The STIRPAT Model

The original IPAT (Impact of Population, Affluence, and Technology) model of the environment was initially proposed by Ehrlich and Holdren in 1971 [79] which had been widely adopted to quantitatively describe the impact of humanistic driving forces on the environment. However, the IPAT model cannot analyze the degree of influence of factor changes on the environment, and it also limits other factors that affect the environment. Later, Dietz and Rosa (1994) extended it to a nonlinear stochastic regression entitled the STIRPAT model [2], which was shown in Equation (12). The STIRPAT model combined the environmental impact pressure (*I*), population size (*P*), affluence (*A*) and technology level (*T*) to quantitatively measure the impact of relevant factors on environmental changes in a stochastic regression.
(12)$$I=\alpha P^{a}A^{b}T^{c}e$$

Herein, *I* is the environmental impact, α is the model coefficient, *P* represents the population size, *A* represents the affluence, and *T* represents the technical level. Concurrently, a, b and c represent the elasticity coefficient of each influencing factor; e represents the random error, which is used to represent the model other factors. When α = a=b = c=1, the STIRPAT model is equal to the IPAT model. Taking the logarithm of both sides of the equation (12) at the same time, the linear expression was obtained here, as showed in equation (13). The STIRPAT model was established through the index system in Equations (13) and (14) and Table 4.
(13)lnI=lnα+alnP+blnA+clnT+lne
(14)lnI=lnα+alnU+blnA+clnT+dlnS+lne

#### 2.4.2. Unit Root Test and Co-Integration Test

In order to avoid spurious regression in the models, it was necessary to perform a unit root test on the data of two time series to ensure that all the data were in stationary. In the unit root test, the ADF (Augmented Dickey Fuller) method was used to exert the unit root test of each variable. The co-integration test is usually adopted to explore whether there was a co-integration relationship between different variables. Herein, the EG two-step method was used to test for the co-integration between the data of two time series.

The specific formulas of ADF test and co-integration test of EG two-step method were shown in Equations (15) and (16).
(15)$$\ln I=\ln\alpha +\mathrm{aln}U+b\ln A+c\ln T+d\ln S+p\ln U^{2}+\mathrm{lne}$$
(16)$$\ln I=\ln\alpha +\mathrm{aln}U+b\ln A+c\ln T+d\ln S+q\ln A^{2}+\mathrm{lne}$$

#### 2.4.3. Detection of Data Collinearity

In a multiple regression model, if two or more explanatory variables are highly correlated, it will lead to the variance expansion of the regression model and parameter estimation, making the economic meaning of the parameter estimator unreasonable. Thus, the predictive function of the regression model will fail. Taking carbon emission as the explained variable, with an urbanization rate, per capita GDP, energy intensity and industrial structure as the explanatory variables, a least squares regression analysis was conducted to detect the data collinearity with variance variations.

#### 2.4.4. Ridge Regression

Ridge regression is a regression model specially designed to solve collinear data analysis and it is still an improved least squares estimation method used here [80]. By abandoning the unbiased estimation of least squares, a more realistic improved ridge regression model was obtained at the cost of losing some data and accuracy. The study employed the Ridge Regression Program in the IBM SPSS Statistics software version 21 for Windows^®^ (SPSS Inc., Chicago, USA.) to obtain the results of the parameters selected for ridge regression. The prediction equation of ridge regression model is shown in Equation (17).
(17)lnI=0.975−0.447lnU−0.187lnA+0.330lnT−0.156lnS

#### 2.4.5. The Environmental Kuznets Curve

Grossman and Krueger (1995) proposed an inverted “U”-shaped correlation between economic income and environment through data [81]. They named it as environmental Kuznets Curve or EKC in abbreviation [81,82,83,84,85,86,87,88,89]. Environmental Kuznets Curve reflects that environmental pollution will rise with economic growth in the early stage and decrease in the later stage in the social economic development. Recently, the environmental Kuznets curve is often adopted by the scholars to explore the correlation between economic development and environmental pollution and governance [81,82,83,84,85,86,87,88,89] to judge whether there is a pattern of pollution first and then treatment in the regional economic development with carbon consumption and emissions from carbon sources. In the study, carbon sources refer to the parent bodies that release carbon dioxide into the atmosphere in nature, and carbon sink refers to the natural host body of carbon or the forests’ capacity to absorb and store carbon dioxide. The reduction of carbon sources is generally achieved through carbon emission reduction, while the technology of carbon sequestration is mainly used to increase the carbon sink sources. Carbon sequestration is also called the carbon sink project, which refers to measures to increase the carbon content of carbon pools other than the atmosphere, including physical carbon sequestration and biological carbon sequestration. In December 1997, in order to mitigate the trend of global warming, representatives of 149 countries and regions adopted the so-called Kyoto Protocol in Kyoto, Japan, which officially entered into force on 16 February 2005. As a result, the international “carbon emission trading system” (i.e., carbon sink project) has been formed. The Kyoto Protocol, which aims to reduce global greenhouse gas emissions, is an international law of carbon sink project that limits carbon dioxide emissions of all countries in the world. Furthermore, the special relationship between the state’s income and the deterioration of national environmental quality in a country will show an inverted U-shaped curve, according to the hypothesis of the environmental Kuznets Curve (i.e. EKC) [81,82,83,84,85,86,87,88,89]. In other words, in the early stage of economic development, the environmental quality deteriorated slightly, but with the increase in national income, the environmental quality will gradually deteriorate. However, when the regional economic development reaches a turning point, the industrial structure gradually transforms from manufacturing to service or technology intensive industries, and the environmental quality will be improved. Though it has not been empirically proved that sustained economic development can automatically improve the deteriorating environmental quality, these regions with the most serious deterioration cases of environmental quality are usually developing countries with the rapid economic growth. Meanwhile, the developed countries have a better environmental quality.

Concurrently, there are many researchers analyzed and reported the impact of social urbanization on the regional carbon emissions to revisit the environmental Kuznets curve in China [90,91,92]. An additional square term of GDP per capita variable is added into the STIRPAT regression model defined above to explore whether there is an environmental Kuznets curve on the carbon emissions in Anhui Province. Based on the result of the STIRPAT model, the study expanded the relationship between the carbon emission, environmental Kuznets curve and urbanization in Anhui Province. It further refined the changing law of carbon emissions in Anhui Province, making the recommendations more targeted.

In accord with the environmental Kuznets curve, when economic development enters an advanced stage, environmental pollution will decline with economic growth. Herein, the square terms of per capita GDP and urbanization rate were evaluated and explored to test whether there is a pattern of the environmental Kuznets curve and to explore the impact of urbanization on the carbon emission in Anhui Province.

#### 2.4.6. The SYS-GMM (System Generalized Method of Moment) Estimator Method

In general, the error term of panel data regression model is composed of two parts. One is related to individual observation unit. It summarizes all the factors that affect the explained variables, but do not change with time. The other summarizes the unobservable factors that change with time due to the section, which are usually called the specific error or the specific disturbance term error (in fact, the second part of the error can also be divided into two sub-parts again). If the panel data regression model is endogenous, the explanatory variable related to the error item is defined as an “endogenous variable”. On the contrary, if it is unrelated to the error term, it is named an “exogenous variable”. The serious consequence of endogenous variables is that the estimator of the regression model will be inconsistent. In the case, there is the so-called endogeneity in the regression model with endogenous variables. The endogeneity indicates one or more explanatory variables are related to the random disturbance term in a given panel data regression model. The explanatory variable and the explained variable interact, influence and cause each other in the model. Thus, no matter how large the sample size is, the estimator of the regression model will not converge to the true parameter value. In econometrics, all the explanatory variables related to the disturbance error terms may be called “endogenous variables” due to the following endogenous sources: (1) the issue of missing variable deviation, (2) the classical measurement error problem, and (3) the simultaneous reverse causality.

At the same time, this study further employs the method of SYS-GMM (System Generalized Method of Moment) estimator to test and delete the possible endogenous effects among variables [93]. The SYS-GMM estimator method solves some missing errors in differential GMM [93]. By adding the lag term of the explained variables, the study establishes the specific measurement model based on the SYS-GMM estimation method in Equation (18). In fact, the SYS-GMM estimator method is applicable regardless of whether there are endogenous explanatory variables in the regression model.
(18)$$\ln I_{t}=\ln\alpha +\chi \ln I_{t-1}+\mathrm{aln}U+\mathrm{bln}A+\mathrm{cln}T+\mathrm{dln}S+\mathrm{qln}A^{2}+\mathrm{lne}$$

## 3. Results

### 3.1. The Carbon Emission Inventory of Anhui Province

At present, there are no statistics and details of urban residents’ food consumption in the released versions of Anhui Statistical Yearbooks. It was assumed that the urban residents and rural residents had similar consumption structures in order to ensure the research’s integrity.

The specific calculation formula was set as follows, i.e. urban residents’ food consumption carbon emissions = number of urban residents × rural residents’ per capita food consumption carbon emissions × urban residents’ per capita food consumption/rural residents’ per capita food consumption. A few missing values are handled by previously described interpolation methods [77,78]. Carbon emission inventory in Anhui Province is shown in Table 5.

Herein, it was found that carbon emissions have grown rapidly in Anhui Province, from 44.2672 million tons in 2000 to 110.2791 million tons in 2019, with an increase of 249% and an annual growth rate of 4.92% (Table 5). Among them, the proportion of carbon emissions from energy consumption increased from 77.29% in 2000 to 91.06% in 2019, which is one of the primary sources of carbon emissions. Food consumption carbon emissions, waste carbon emissions, cultivated land carbon emissions and ruminant carbon emissions followed closely, which together constituted the source of carbon emissions in Anhui Province. The carbon sink capacity of Anhui Province increased from 62.0763 million tons in 2000 to 83.7051 million tons in 2019 (Table 5), with an increase of 135% and the annual growth rate of 1.59%, only 1/3 of the growth rate of carbon emissions. Before 2009, the discharge amount of carbon emissions was less than the maximum capacity of carbon absorption in Anhui Province. However, it surpassed the maximum capacity of provincial carbon absorption for the first time in 2010. This indicated that carbon emissions have grown rapidly and have not been effectively absorbed with the rapid economic development and industrialization of Anhui Province.

Carbon emissions caused by energy consumption are the primary sources of carbon emissions in Anhui Province. In particular, the energy structure of Anhui Province has undergone the major adjustments, whereas the carbon emissions of various industries were significantly different which was explored and analyzed as follows.

Firstly, the carbon emissions caused by energy consumption increased from 34.2158 million tons in 2000 to 100.424 million tons in 2019, an increase of 294% and an annual growth rate of 5.83%. Among them, the proportion of carbon emissions caused by coal, oil and electricity consumption in 2000 was 78.5%, 6.9% and 11.3%, respectively. By 2019, the proportion of its energy structure was modified to 38.8%, 19.5% and 31.1%. It could be found that although coal was still the main body of fossil energy consumption, due to the large-scale use of oil and electricity, the energy structure originally dominated by coal has gradually shifted to the current coordinated development of coal, oil and electricity.

Secondly, there were large differences in the carbon emissions among different industries. Production carbon emissions increased from 31.6109 million tons in 2000 to 94.2356 million tons in 2019, an increase of 298% and an annual growth rate of 5.92%. Among them, the carbon emissions of the primary industry increased from 798,000 tons in 2000 to 1,253,000 tons in 2019, with an annual growth rate of 2.4%. The ratio of carbon emissions to productive carbon emissions remained at around 0.01, which was relatively stable. The carbon emissions of the secondary industry increased from 29.3025 million tons in 2000 to 82.9985 million tons in 2019, with an annual growth rate of 5.6%, which is the main target of carbon emissions. The carbon emissions of the tertiary industry increased from 1.5124 million tons in 2000 to 9.9841 million tons in 2019, with an annual growth rate of 10.4% and the fastest growth rate of carbon emissions.

Thirdly, the living carbon emissions grew rapidly, and there was a significant difference between the energy consumption levels of rural and urban areas. Living carbon emissions increased from 2.6049 million tons in 2000 to 6.1884 million tons in 2019, an increase of 169% and an annual growth rate of 4.67%. Among them, urban and rural living carbon emissions increased from 1,993,400 and 611,500 tons in 2000 to 3,367,600 and 2,820,800 tons in 2019, with an annual growth rate of 2.8% and 8.4%, respectively. It can be found that although urban living carbon emissions are high, however, the growth rate of residential carbon emissions in rural areas is much higher than that in urban areas. In terms of energy consumption types, the proportion of carbon emissions caused by coal consumption in urban and rural areas in 2000 was 84.8% and 100%, respectively, but later dropped to 2.3% and 3.9% in 2019. At the same time, carbon emissions from oil and electricity have grown rapidly. From 2012 to 2019, the annual growth rate of carbon emissions from oil consumption in urban and rural areas was 20.5% and 40%, respectively, and the annual growth rate of carbon emissions from electricity consumption was 7.25% and 9.1%.

Finally, it can be found that the consumption of coal and electricity in rural areas is higher than that in urban areas by comparison, whereas the consumption of oil in rural areas is lower than that in urban areas. This is due to the different energy consumption structures in rural and urban areas. Rural areas mainly consume coal and biomass energy, while cities mainly consume natural gas and oil; towns have relatively sound and convenient transportation infrastructure, resulting in higher oil consumption than rural areas. With the change of the energy structure, the source of carbon emissions has gradually changed from a single coal to an equal emphasis on coal, oil, and electricity. It can be seen that reducing carbon emissions should continue to reduce the consumption proportion of coal and improve the efficiency of coal use to promote the gradual shift of oil consumption to clean energy consumption. In view of the different energy consumption structures in urban and rural areas, it is necessary to promote the transformation and development of urban areas towards clean energy, advocate green travel, and develop new energy vehicles. The government should continue to promote the level of urbanization development and improve rural infrastructure construction.

Carbon emissions of urban food consumption continued to increase from 2000 to 2008. It reached a peak in 2008. Next, it began to decline after 2009, and then remained in a relatively stable state. Carbon emission of rural food consumption had been declining since 2000, and there was a large gap of carbon emissions in food consumption between the urban and rural areas. The urbanization rate of Anhui Province went from 0.28 in 2000 to 0.56 in 2019, with an annual growth rate of 3.67%. With the continuous implementation of urbanization, the rural population continues to decline, which is the crucial reason for the continuous decline of residents’ carbon emissions from food consumption in rural areas.

From the perspective of residents’ food consumption structure, the per capita consumption of grain and sugar with high traditional carbon emission coefficients dropped from 270.24 kg and 1.59 kg in 2000 to 169.2 kg and 1.1 kg in 2019, with an annual decline rate of 2.43% and 1.92%, respectively. Vegetables, pork, poultry, eggs, and fish increased from 80.28, 9.51, 3.57, 5.56 and 3.5 kg in 2000 to 99.3, 22.1, 14.5, 12.1 and 13.2 kg, respectively. The consumption of pork, poultry, eggs and fish has increased significantly, replacing the consumption of foods with high carbon emission coefficients such as grain, and reducing carbon emissions. The future dietary structure of residents should develop towards vegetables and fruit with lower carbon emission coefficients, and advocate for a green diet.

The carbon emissions caused by waste water remained at a relatively stable level, while the carbon emissions caused by garbage disposal increased rapidly. From 2000 to 2008, carbon emissions from wastewater increased from 64,100 tons to 178,700 tons, an annual growth rate of 13.7%. Since that, government of Anhui Province strengthened the purification and treatment of wastewater, which dropped from 178,700 tons in 2008 to 81,000 tons in 2019. Carbon emissions from solid waste have grown rapidly, from 77,600 tons in 2000 to 2.147 million tons in 2019, with an annual growth rate of 19.1%. Among them, in 2019, the carbon emission caused by waste incineration was 27.6 times that of the landfill, and it was the main source of solid waste carbon emissions. Carbon emissions from waste incineration increased from 3,600 tons in 2003 to 2,072,000 tons in 2019, with an annual growth rate of 48.8%. Carbon emissions from landfills increased from 77,600 tons in 2000 to 165,100 tons in 2013, an annual growth rate of 6%. However, with the development of landfill technology, the carbon emissions caused by landfill decreased from 165,100 tons in 2013 to 75,000 tons in 2019, with an annual decline rate of 12.3%. It can be seen that in terms of waste carbon emissions. Anhui Province has relatively effective controls on waste water and landfill, and the carbon emissions caused by waste combustion are the key to reducing emissions.

The carbon sink capacity has been gradually enhanced in Anhui Province, in which crops played an important role. The carbon sink capacity increased from 62.076 million tons in 2000 to 83.705 million tons in 2019, with an annual growth rate of 1.59%. Among them, forest carbon absorption increased from 12.6429 million tons in 2000 to 15.0803 million tons in 2019, with an annual growth rate of 0.93%. Since the grassland area in Anhui Province has not changed much, its carbon absorption has been stable at 1.5771 million tons. Crops account for the largest amount of carbon absorption, increasing from 47.856 million tons in 2000 to 67.048 million tons in 2019, an annual growth rate of 1.8%. It is found that the carbon sink capacity is mainly enhanced due to the continuous closure of mountains and forests in Anhui Province, with the continuous expansion of the forest area and changes of crop planting structure.

### 3.2. Carbon Emissions Per Capita and Carbon Intensity

In addition to the total carbon emissions, industrial sectors’ carbon emission intensity and per capita carbon emissions were also the important indicators to measure the level of regional carbon emissions. Among them, carbon emission intensity was the ratio of carbon emission to GDP, and per capita carbon emission was the ratio of carbon emission to the indigenous population. The per capita carbon emissions of the primary, secondary and tertiary industries were the ratios of industrial carbon emissions to the number of employees in the industry, respectively. The carbon emission intensities of primary, secondary and tertiary production were the ratio of industrial carbon emissions to industrial production values (GDPs), respectively. This study set 2000 as the base period and deflated the GDP from 2001 to 2019 with the details showed in Table 6.

The per capita carbon emissions in Anhui Province showed an increasing trend, and the carbon emission showed a downward trend. Carbon emissions per capita increased from 0.727 tons in 2000 to 1.732 tons in 2019 (Table 6), with an annual growth rate of 4.68%. Among them, the per capita carbon emissions of primary, secondary and tertiary industries increased from 0.039, 5.011, and 0.179 tons in 2000 to 0.093, 6.582, and 0.562 tons in 2019, with the annual growth rates of 4.6%, 1.4%, and 6.2%, respectively (Table 6). In brief, the per capita carbon emissions from the tertiary industry are growing rapidly. During the period from 2000 to 2019, the annual decline rate of the number of employees in the primary industry was 2.1%, and the annual growth rate of the number of employees in the secondary and tertiary industries was 4.13% and 3.97%, respectively (Table 6). The number of industrial employees grew as the fastest. By comparison, it was found that the ranking of measured data were secondary industry > tertiary industry > primary industry in terms of per capita carbon emissions. Simultaneously, in terms of the growth rate, the measured data were ranked as tertiary industry > primary industry > secondary industry. General carbon emission of Anhui Province decreased from 1.41 tons of carbon yuan^−1^ in 2000 to 0.44 tons of carbon yuan^−1^ in 2019 (Table 6), with an annual decline rate of 5.9%.

In addition, the carbon emission of the primary, secondary and tertiary industries decreased from 0.107, 2.759, 0.114 tons of carbon ten thousand yuan^−1^ to 0.064, 0.808, 0.079 tons of carbon ten thousand yuan^−1^ in 2019, while the annual decline rates were 2.7%, 6.2%, and 1.9%, respectively (Table 6). It can be noted that the carbon emission of the secondary industry has dropped the fastest (Table 6). Further promoting the adjustment of industrial structure and vigorously developing the industrial sectors of services can effectively reduce the carbon emission of Anhui Province.

### 3.3. Unit Root Test and Co-Integration Test

In the study, it was necessary to do a unit root test on the data of two time series to guarantee that all the data were in stationary and prevent any spurious regression in the subsequent applied models. First, the ADF method was used to test the unit root of each variable. It was found that these five variables were all non-stationary sequences that became stationary sequences after the first-order difference. Next, using the EG two-step method to explore whether there was a co-integration relationship between different variables. It is found that the estimated residual sequence of the regression formula was a stationary sequence when the original data were used for regression analysis. Therefore, there was a long-term co-integration relationship between carbon emission and urbanization rate, per capita GDP, energy intensity and industrial structure, indicating that regression analysis could be conducted. The analyzed results are summarized and showed in Table 7.

#### 3.3.1. Detection of Data Collinearity

Before exerting the regression models in the study, least squares regression analysis was conducted to avoid the high correlation among explanatory variables. Ultimately, the expansion of the variance was prevented in the least squares regression and its parameter estimation, making the economic meaning of the parameter estimator reasonable. The provincial carbon emission was taken as the explained variable, while urbanization rate, per capita GDP, energy intensity and industrial structure were taken as the explanatory variables, the least squares regression analysis was conducted. It was found that the variance inflation factor VIF of most explanatory variables was greater than the critical value of 10, indicating a high degree of collinearity among the explanatory variables.

#### 3.3.2. Ridge Regression

With the special SPSS Ridge Regression Program, the ridge regression model was designed and exerted to solve the problem of multiple collinear data analysis and obtain the parameters screened for ridge regression. It was found that when k = 0.4, variation of each coefficient tended to be stable and the corresponding R^2^ = 0.975, which achieved a better fitting effect than the original ridge regression. The ridge regression model F statistic was 145.1, *p* value equaled to 0.00, and the test was significant in the study. By bringing the panel data into equation 15, we could calculate the provincial carbon emissions and compare it with the actual carbon emission of Anhui Province from 2000 to 2019 (Figure 1). Through comparison, it was found that the provincial carbon emissions were decreasing year by year in Anhui Province, with an average estimation error of 4.5% which indicated the predicting regression model had a high accuracy.

### 3.4. Scenario Analysis and Prediction of the Changes of Carbon Emissions Based on the STIRPAT Model

#### 3.4.1. Scenario Analysis and Parameters Set for Various Driving Factors on Social Economic Development in Anhui Province

In this section, scenario analysis and prediction were conducted for the changes of carbon emissions based on the STIRPAT model defined in Equation (14) and Table 4. In order to avoid the multicollinearity interference of the panel data, this study uses the ridge regression method to regress the data to retain the information of the independent variables and dependent variables to a greater extent. Referring to the social and economic development trends and future developing plans of Anhui Province, the study designed and set the parameters for four driving factors of social development in Anhui Province, i.e., urbanization rate, per capita GDP, energy intensity and industrial structure. In the parameter designing and setting, the impact of the new crown epidemic and emerging industries were considered in the social economic development of Anhui Province. Subsequently, there were three scenarios ranked with the parameters set, i.e., fast, medium and slow (Figure 1 and Table 8). Combined with the “14th Five-Year Plan” and the Vision Outline of the year 2035 for Anhui Province, the change rate of each variable for these driving factors under different scenarios was adjusted in a five-year cycle in Anhui Province.

Scenario setting of the urbanization rate (U): From 2000 to 2019, the urbanization rate in Anhui Province increased from 0.28 to 0.56, with an average annual growth rate of 3.7%. In the 14th Five-Year Plan of Anhui Province, the urbanization rate would reach 62%. Based on this, the annual growth rate of urbanization in Anhui Province would reach 1.8%, so the annual growth rate of medium-speed urbanization in 2020–2025 was set to 1.8%. The growth rate has fallen by 0.1% every five years thereafter. Growth rates are set to 2% and 1% in the fast and slow scenarios, respectively.

Scenario setting of per capita GDP (A): From 2000 to 2019, the per capita GDP of Anhui Province increased from 5,129 yuan to 26,133 yuan, with an annual growth rate of 8.9%. In the 14th Five-Year Plan of Anhui Province, the annual growth rate of per capita GDP was set at 5.6%. Based on this, the medium-speed per capita GDP annual growth rate from 2020 to 2025 was set at 5.6%, followed by a drop of 1% every five years. Growth rates were set at 6% and 4.6% in the fast and slow scenarios, respectively.

Scenario setting of energy intensity (T): The energy intensity of Anhui Province decreased from 1.56 in 2000 to 0.56 in 2019, with an annual decline rate of 5.3%. Among them, the annual decline rate of energy intensity during the 13th Five-Year Plan period was 6.3%. Based on this, the annual rate of decline in medium-speed energy intensity was set at 6.5% for 2020–2025, followed by 0.1% every five years thereafter. The annual rates of decline in energy intensity in the fast and slow scenarios were 7% and 5%, respectively.

Scenario setting of industrial structure (S): It could be observed that the industrial structure increased from 0.27 in 2000 to 0.42 in 2012 in Anhui Province, with an annual growth rate of 6.7%. It peaked in 2012 before falling to 0.31 in 2019. Since Anhui Province was a major manufacturing province, in the 14th Five-Year Plan, the proportion of manufacturing value added to GDP would increase from 26.3% in 2020 to 30% in 2025. Based on this, the annual growth rate of the medium-speed industrial structure from 2020 to 2025 was set at 2.6%, followed by a drop of 1% every five years. The annual growth rates of industrial structure in the fast and slow scenarios were set at 3% and 1.6%, respectively. These specific parameter settings are shown in Table 8.

#### 3.4.2. Scenario Prediction on Social Economic Development in Anhui Province

Carbon emission is an important indicator to measure the level of regional carbon emission. With reference to the “14th Five-Year Plan” of Anhui Province and the Outline of the Vision for the year 2035, the annual growth rate of Anhui Province’s GDP was set at 6%, and the GDP of Anhui Province is predicted for the period 2020–2040. In the official document “Strengthening Action on Climate Change—China’s Nationally Determined Contribution”, China promised to reduce carbon emission by 60–65% in 2030 compared with 2005. This study used these references as the standard to compare the changes in carbon emission under different scenarios, as well as to analyze its emission reduction effect. The specific results are shown in Figure 2.

Carbon emissions in the three ranked scenarios were in the following sequence as fast development mode < medium-speed development mode < slow-speed development mode. The carbon emission of the slow, medium and fast development models will drop from 0.49 in 2020 to 0.27, 0.21 and 0.19 in 2040, with an annual decline rate of 3%, 4% and 5%, all lower than 5.9% in 2000–2019. The carbon emissions of the slow, medium and fast development models will increase from 12,988, 12,833, and 127.82 million tons in 2020 to 22,547, 17,548, and 161.49 million tons in 2040, with annual growth rates of 2.8%, 1.6%, and 1.2%, respectively. All these data are lower than the average annual growth rate of carbon emissions (4.9%) in Anhui Province from 2000 to 2020. The carbon emission of Anhui Province in 2030 is expected to drop by 65% compared with 2005, and the specific value is 0.35. By comparison, it was found that the carbon emission intensities in 2030 would be 0.36, 0.31 and 0.30 under the slow, medium and fast development modes, respectively. It could also be found that the slow development mode did not meet the expected requirements in Anhui Province.

### 3.5. Revisiting The Environmental Kuznets Curve

According to the environmental Kuznets curve, when economic development enters an advanced stage, environmental pollution will decline with economic growth. Since the inverted “U”-shaped correlation between the resident income and environment was proposed as environmental Kuznets Curve in 1995 [79,80,81], more and more researchers explored and analyzed the environmental Kuznets curve in different areas to evaluate the correlation between economic development and environmental pollution and governance [80,81,82,83,84,85,86,87,88,89,90,91,92] to judge whether there is a pattern of pollution first and then treatment in regional economic development. Based on the STIRPAT model, this study evaluated the possible relationship between carbon emission, environmental Kuznets curve and urbanization in Anhui Province to further explore the changing law of carbon emissions in Anhui Province with the targeted recommendations.

Based on the results of the STIRPAT model and ADF tests, the square terms of per capita GDP and urbanization rate were adopted to test whether there is an environmental Kuznets curve in Anhui Province and the impact of urbanization on the carbon emissions of Anhui Province. The specific formulas were shown in equations 16–17. In order to ensure the accuracy of the results, ADF tests were conducted on the square term of per capita GDP and urbanization rate. It was found as a non-stationary sequence, but the first-order difference was a stationary sequence. The formula was further tested for co-integration using the EG two-step method and found to pass the linear co-integration test at the 10% level.

In order to ensure the accuracy of the regression results and avoid the influence of collinearity, this study still uses ridge regression for analyzing the correlation. The regression results show that the model fitting coefficient R^2^ is 0.98, which is a good fitting effect. Except for the industrial structure, other original variables all passed the test at the 0.01 significant level. The coefficients of per capita GDP correlated with carbon emissions and its squared term are −0.148 and −0.008, respectively, and the correlation curve is in an inverted “U”-shaped curve, in line with the environmental Kuznets curve. This is an approximate inverted “U”-shaped Kuznets curve between carbon emission and economic development in Anhui Province. Furthermore, the coefficients of the urbanization rate and its square term are −0.353 and 0.179, respectively. However, the curve of the correlation between carbon emission and urbanization rate showed a positive “U” shape, which has economic implications too. The latter “positive U”-shaped correlation showed the great emission reduction potential with carbon sink under the environmental Kuznets curve in developing areas in Anhui Province, China. The possible reasons will be explored later with the well-known environmental Kuznets curve.

### 3.6. The Result of SYS-GMM Estimator Method

Based on the STIRPAT regression and the other analyses, the square term of GDP per capita variable is added to test whether there is an environmental Kuznets curve revisited later on carbon emissions in Anhui Province. In order to exclude the possible emerging endogeneity of panel data regression model to ensure the predicting accuracy of the regression model, the study conducted an ADF test on the square term of per capita GDP variable. It was found that there was a non-stationary series and the corresponding first-order difference is a stationary series. At the same time, this study used the EG two-step method to conduct co-integration tests on the equation, and found that it passes the linear co-integration test at the 10% level. In order to exclude the possible endogenous effects among later added variables, the SYS-GMM estimator method was further adopted to detect all the possible endogenous variables. Then, a new regression model was established by adding the lag term of the explained variables in Equation (18) above. Subsequently, the study analyzed and explored it based on the SYS-GMM estimator method. The results of the new regression model showed that the lag coefficient of lnI was 0.366, which was significant at the level of 1%. The estimated *p* values corresponding to the lag term AR (1) and AR (2) are 0.085 and 0.553, respectively. Meanwhile, the *p* value of theSargan test was 0.685, which indicated that the econometric regression model had a dynamic relationship and can be estimated using the system GMM method. The lagging term of carbon emission intensity was estimated as 0.366 at the significant level of 1%, indicating that carbon emission intensity was positively affected by the previous study period. The carbon emission intensity of the previous period has an inertial effect on the investigated years. Results showed that the primary variables, except energy intensity, passed the test at the significant level of 1%. The coefficients of per capita GDP and its square term were 1.476 and −0.12, respectively. Thus, it showed an inverted “U” curve too, which indicated that there was an environmental Kuznets curve indeed for the carbon emissions in Anhui Province. Furthermore, the first term coefficient of per capita GDP was positive, indicating that the current economic development has promoted the carbon emission intensity in Anhui Province. However, the environmental relationship curve between the carbon emission intensity and the economic development rate has not yet reached the key inflection point of environmental Kuznets curve (IPEKC). Therefore, the government might strengthen environmental governance and actively promote industrial transformation, as well as increase the proportion of clean energy in the industrial production structure.

## 4. Discussion

At present, China’s pledge is an ambitious target of carbon emissions as a rapidly developing country. The country previously pledged to turn around the constant growth in its carbon emissions by 2030 at the latest and increase the fraction of its forthcoming energy from the zero-carbon sources to 20% by the same year [13,14,15]. However, the commitment raises serious questions in developing areas such as Anhui Province in China: Are these development goals realistic? A theoretical analysis would produce insights into how such policies might affect China’s energy system and carbon emissions. It might shed light on the level of carbon tax that might be required to achieve a given emissions reduction. This information could help guide Chinese policy makers as they further define the details of their plans. Therefore, the current study was designed to estimate and evaluate the emissions and growth rates of each carbon emission sources by compiling the carbon emission inventory in Anhui Province from 2000 to 2019 with those above mentioned five defined sources of carbon emissions. It was also aimed to compare and explore whether there was an environmental Kuznets curve occurring in Anhui Province. The analyzed results show that the models had a good stability in model fitting and the time series data were in stationary in the unit root and co-integration tests. Specific analyses of the ADF test and the co-integration test of the EG two-step method found that the estimated residual sequence of the regression formula was a stationary sequence when the original data were used for the regression analysis. Therefore, there was a long-term co-integration relationship between the adopted variables, and the regression models were well conducted without multicollinearity interference by detecting data collinearity. Thus, the study provides an empirical basis for the advance development research on carbon emission characteristics and reduction potential and low-carbon emission reduction policy-making in developing areas such as Anhui Province in China.

The study differs from the other research reports and their achievements devoted to low carbon emissions in the development characteristics and regional reduction potential of carbon emissions and carbon sinks in the developed areas in China [17,18,19,20,21,22,23,24,94,95,96]. For instance, the total carbon emissions increased sharply, from 76.11 to 140.19 TgC yr^−1^ at the provincial level, with an average annual growth rate of 10.52%, while the vegetation carbon sinks declined slightly, from 54.52 to 53.20 TgC yr^−1^ in Guangdong Province [17]. Similar case was reported that carbon emissions of forest ecological products showed a fluctuating upward trend in Zhejiang Province [94] and Xinjiang Province [95], whereas it was regarded that urgent measures should be taken to significantly reduce greenhouse gas emissions in Shandong Province, a typical province of energy consumption [19]. However, there are also some cases that reported the characteristics and dynamics of carbon emissions and carbon sinks with emission reduction potential insights in economically developing areas of China [40,75,77]. These reports found that there was much reduction potential of carbon emissions with the carbon sink as well as carbon emissions per capita and carbon intensity in Chinese economically undeveloped and developing areas [40,75,77]. They also proposed that the government should provide a suitable economic growth and development range for the average gross domestic product growth rate, the proportions of primary industry and secondary industry, energy consumption intensity of secondary industry, and the urbanization [40,75,77]. In particular, there was a similar inverted “U”-shaped industrial influence mechanism of rural financial scale and incorporating rural financial efficiency and agricultural carbon emissions with stringent carbon development scenarios of the environmental Kuznets curve in Henan Province, China [40]. In this study, resulted coefficients of per capita GDP correlated with carbon emissions and its squared term are −0.148 and −0.008, respectively, and the correlation curve is in an inverted “U”-shaped curve, in line with the environmental Kuznets curve. This is an approximate inverted “U”-shaped Kuznets curve between carbon emission and economic development in Anhui Province. Furthermore, the coefficients of the urbanization rate and its square term are −0.353 and 0.179, respectively. However, the curve of the correlation between carbon emission and urbanization rate showed a positive “U” shape, which has economic implications too. The latter “positive U”-shaped correlation showed the great emission reduction potential with carbon sink under the environmental Kuznets curve in developing areas in Anhui Province, China. The reasons for these changes are properly that all cities and industrial districts in Anhui Province are promoting the new energy vehicles and industrial green technologies, whereas the provincial industrial and agricultural enterprises are following a national sustainable development pathway. However, the existing premise of the conventional environmental Kuznets curve is that most regional economics follows the development pathway of pollution first and then treatment [81]. This can be found from the changing laws of wastewater in the above carbon emission inventory and many other government documents in Anhui Province. The advancement of the urbanization rate has increased the agglomeration effect and scale effect of low-carbon technologies in Anhui Province, which has suppressed carbon emissions in Anhui Province to a certain extent. However, with the rapid development of urbanization in Anhui Province, many related issues such as whether the spatial distribution is reasonable and whether building materials are energy-saving will gradually be exposed. This requires the government of Anhui Province to further implement energy-saving and emission-reduction measures to avoid the provincial emergence of inflection points. Under the “14th Five-Year Plan”, it is a major mission for the government to reduce its carbon emission to 65% of that in 2005 by 2030 in Anhui Province. It was suggested to implement relevant low-carbon policies to avoid inflection points of the well-known environmental Kuznets curve.

The used models and methods of this study are equally applicable to other regions and researches and/or investigations. For example, an improved STIRPAT model was adopted to explore the critical influencing factors of carbon emission in the Yangtze River Delta [67], whereas an extended STIRPAT model was employed to analyze the multivariate driving factors of carbon emissions from energy consumption in Xinjiang Province [95] and the other provincial household carbon emissions in China [96]. Other research reports employing similar models and methods are found in the web search too.

In short, these suggested a great potential of emission reduction with carbon sink in Anhui Province. Concurrently, the local government should strengthen the environmental governance, actively promote industrial transformation, and increase the proportion of clean energy in the energy production and consumption structures in Anhui Province. Therefore, this study provides new insight for the government to achieve carbon reduction goals and realize low-carbon development in Anhui Province.

## 5. Conclusions

The study empirically compared and evaluated the carbon emissions and growth rates of each carbon emission sources by compiling the carbon emission inventory in Anhui Province with the five defined sources of carbon emissions collected from 2000 to 2019. Next, the STIRPAT model was used to analyze the carbon emission reduction potential of Anhui Province, and explore the changes of carbon emission in Anhui Province under different scenarios. The study found that carbon emissions in Anhui Province increased rapidly from 44.2672 million tons in 2000 to 110.2791 million tons in 2019, with an annual growth rate of 4.92% during the period of 2000–2019. Carbon emission caused by energy consumption was identified as the primary source of carbon emission in Anhui Province, while the energy structure was gradually adjusted from the initial single coal consumption to the coordinated development of coal, oil, electricity and other energy sources. There were big differences in the carbon emissions among different industries. In terms of carbon emissions, the ranking of measured data was as secondary industry > tertiary industry > primary industry. In terms of carbon emission growth rate, the measured data were ranked as tertiary industry > secondary industry > primary industry. The secondary industry (i.e., industry) accounted for the largest proportion in carbon emissions, and the primary industry (i.e., agriculture) was ranked second in carbon emissions. The tertiary industry (i.e., services) had the fastest growth rate of carbon emissions. There were significant differences in energy consumption and energy structure between urban and rural areas. The carbon emission level of urban energy consumption was higher than that of rural areas, but the growth rate of carbon emissions in rural areas was much higher than that in urban areas. Electricity and oil were the main consumptive sources in urban areas, and coal and electricity consumption in rural areas. Carbon emissions from food consumption showed a trend of first increasing and then decreasing in urban areas, and decreasing trends in rural areas. Carbon emissions from waste have increased rapidly. Among them, carbon emission caused by waste incineration was the main sources. The local government should strengthen the environmental governance, actively promote industrial transformation, and increase the proportion of clean energy in the energy production and consumption structures in Anhui Province. Under different scenarios, the reduction of carbon emission in Anhui Province was different. The fast and medium-speed development models might achieve the predetermined social development goals.

## 6. Remarks on the Limitations and Future Directions for Research

The study concentrated on carbon emissions and growth rates of carbon emission sources by compiling the carbon emission inventory in Anhui Province, China. Carbon emission caused by energy consumption was identified as the primary source of carbon emission in Anhui Province, while the energy structure was gradually adjusted from the single coal consumption to the coordinated development of coal, oil, electricity and other energy sources. Nevertheless, public health, green economy and environmental protection are among the intended outcomes of the energy production and consumption initiatives. Accordingly, the governmental decision-making process tends to adopt polycentric policies in environmental governance. We employed the STIRPAT model to analyze the carbon emission reduction potential of Anhui Province, and explore the changes of carbon emission in Anhui Province under different scenarios. Limitations are conversely of importance to improve the group’s research in the future. One of the limitations is the sampled districts and areas at the state level in Anhui Province, whereas there are many similar developing provinces within mainland, such as Jiangxi Province, Henan Province, Hunan Province, and Hubei Province. Future research should take all the opportunities to sample at the national level or a large economic zone such as the Yangtze River delta. The other limitation is that the study did not include some special control variables in the regression equations, including the scales of regional economies and communities, the environmental R&D intensity, and the average weight of the high-tech industries, etc. Usages of such variables may affect the ultimate estimates to obtain more proper parameters and better regression models. Specific, necessary attention to these control variables may be the research focus of future studies. In addition, a more comprehensive empirical analysis is also needed to conduct an investigation on the carbon emission scenarios and the potential for emission reduction in China.

## 7. Policy Recommendations

The government (or governmental sectors) can screen and optimize the regional energy structure and vigorously develop clean energy in the near future. The energy structure of Anhui Province has gradually developed from a single coal resource to a coordinated development of multiple energy sources such as coal, oil and gas and electricity. However, fossil energy is still an important part of primary energy. The government should further optimize the energy structure, continue to reduce the consumption proportion of coal in the energy structure, and vigorously develop clean energy. Aiming at the different energy consumption structure of urban and rural areas, Anhui Province should promote the continuous adjustment of its energy structure. Further strengthen rural infrastructure construction and improve rural transportation facilities. Cities and towns should vigorously develop the clean energy, develop new energy vehicles, and reduce oil consumption.

Furthermore, the government can also adjust and optimize the industrial structure and guide the green development of the tertiary industry while ensuring industrial upgrading. Gradually reducing the proportions of traditional high-carbon emission industries can optimize the ultimate internal industrial structure of the society and vigorously developing high-tech industries. At the same time, the rapid development of the tertiary industry also promotes the continuous upgrading of the industrial structure in Anhui Province, which reduces the carbon emission level of Anhui Province. However, the rapid growth of carbon emissions in the tertiary industry needs attention, and guiding the green development of the tertiary industry is the focus of Anhui Province to achieve emission reduction.

Moreover, the government may constantly improve the garbage disposal mechanism and promote the development of green technology and non-polluting garbage disposal technologies. With the rapid development of cities, carbon emissions caused by waste are equally becoming more and more serious. At present, Anhui Province has relatively effective controls on carbon emissions from sewage and solid waste landfills. However, carbon emissions from solid waste combustion and waste water have grown rapidly and these cases have not been effectively controlled yet. It is necessary to promote the garbage classification and recycling, as well as to improve the garbage classification and recycling system. Subsequently, multiple industrial chains for the pollution-free treatment of industrial solid waste and waste water can be readily constructed, so as to reduce the impacts of waste carbon emissions.

Additionally, the government may well strengthen the construction of carbon sinks and increase the protection of forests and grasslands in Anhui Province. Subdivide the carbon sink area in Anhui Province; delineate protection red lines for major carbon sink sources such as forests, grasslands, and wetlands; and gradually increase the area of artificial afforestation and artificial grassland in Anhui Province. Considering that carbon sinking capacity of crops is the main source of carbon sinks in Anhui Province, ensuring food security is also the responsibility of a province with a large output. This means that it has strong practical significance to include the stablely growing crops as carbon sinks into the carbon neutral development plan of Anhui Province. The government can combine the three strategies of regional green and high-quality development, ecological and livable construction, and food security, as well as explore different development models for emission reduction and increase sinks. It is necessary to seek a balance between production, life and ecology during development and promote the further enhancement of provincial carbon sink capacity in Anhui Province.

Finally, the government may steadily advance and promote the provincial level of urbanization development and optimize its spatial distribution pattern in Anhui Province. The advancement of urbanization has brought economies of scale to the utilization of public goods. At the same time, along with changes in lifestyles and diffusion of technologies, the advancement of urbanization has played a certain role in reducing the intensity of carbon emissions in Anhui Province. On this basis, it is necessary to further optimize the spatial layout of cities and towns to prevent the emergence of inflection points caused by the rapid urban expansion and inadequate implementation of low-carbon measures.

## Figures and Tables

**Figure 1 ijerph-19-16424-f001:**
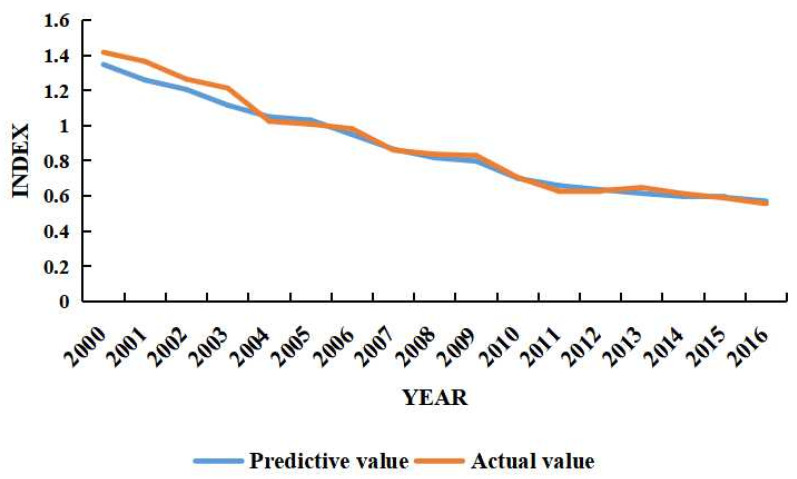
Results of the regression fitting.

**Figure 2 ijerph-19-16424-f002:**
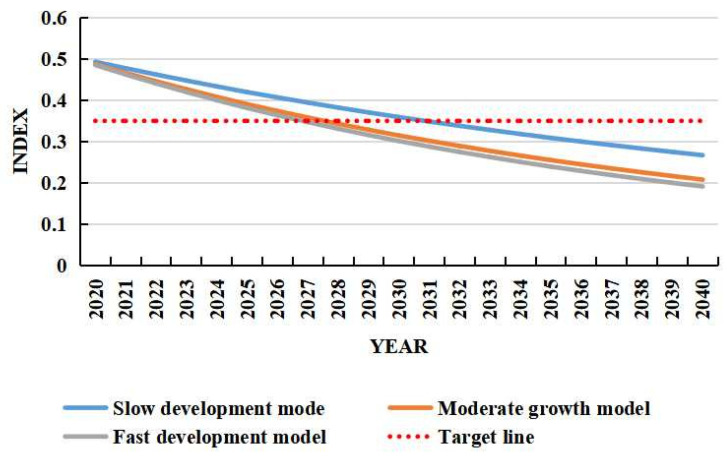
Changes of carbon emission in Anhui Province.

**Table 1 ijerph-19-16424-t001:** Carbon emission coefficient of food consumption (unit: ton carbon/ton).

Type	Grain	Vegetable	Drink	Fruit	Poultry Meat	Aquatic Products	Vegetable Oil	Egg	Sugar	Milk
h	0.3268	0.0274	0.0411	0.0498	0.2546	0.1433	0.7666	0.151	0.3965	0.0629

**Table 2 ijerph-19-16424-t002:** The carbon emission coefficients of ruminant carbons from intestinal peristalsis and fecal (unit: kg/head·year).

Carbon Emission Pathway	Animal Species								
	Cattle	Cows	Horse	Donkey	Mule	Pig	Goat	Sheep	Poultry
Production by enteric fermentation	55	56	18	10	10	1	5	8	0
Production by fecal	0.97	7.95	1.23	0.62	0.62	1.95	0.13	0.1	0.02

**Table 3 ijerph-19-16424-t003:** Crop economic coefficient and carbon absorption rate (unit: ton carbon/ton).

Coefficient	Vegetables	Wheat	Rice	Corn	Potatoes	Soybean	Peanut	Cotton	Rape	Cane	Melons	Tobacco	Other Crops
ω	0.6	0.4	0.45	0.4	0.7	0.34	0.43	0.1	0.25	0.5	0.7	0.55	0.4
S_c_	0.45	0.485	0.414	0.471	0.423	0.45	0.45	0.45	0.45	0.45	0.45	0.45	0.45

**Table 4 ijerph-19-16424-t004:** The established indicators for variables of the STIRPAT model.

Factor	Symbol	Indicator Description	Unit
Carbon intensity	I	The ratio of carbon emissions to GDP	tons of carbon/10,000 yuan
Urbanization rate	U	The ratio of urban population to total population	percentage
GDP per capita	A	The ratio of economic aggregate to population	Yuan/person
Energy intensity	T	Ratio of energy consumption to economic output	tons of standard coal/10,000 yuan
Industrial structure	S	The added value of the secondary industry as a proportion of GDP	percentage

**Table 5 ijerph-19-16424-t005:** The carbon emission inventory of Anhui Province (10,000 tons).

Year	Carbon Source													Carbon Sink			
	Energy Consumption					Food Consumption		Cultivated Land	Ruminants		Waste		Carbon Emissions	Forest	Grassland	Crop	Carbon Uptake
	Productive			Living													
	Primary	Secondary	Tertiary	Town	Rural	Town	Rural		Enteric Fermentation	Stool	Waste Water	Rubbish					
2000	80	2930	151	199	61	486	448	44	9	3	6	8	4427	1264	158	4786	6208
2001	83	3178	160	226	99	503	427	44	8	3	10	7	4746	1208	158	4860	6226
2002	80	3182	177	235	127	513	397	43	8	3	9	6	4780	1208	158	5209	6575
2003	83	3565	194	239	163	531	370	42	8	3	9	6	5214	1208	158	4341	5706
2004	84	3541	226	255	163	484	330	42	9	4	16	10	5156	1265	158	5231	6654
2005	78	3716	254	273	146	504	329	42	9	4	17	8	5373	1372	158	5092	6621
2006	79	4226	284	237	149	549	315	42	9	4	17	10	5913	1372	158	5356	6885
2007	83	4585	328	226	149	516	288	43	9	4	17	13	6254	1372	158	5576	7106
2008	97	5031	363	211	127	715	281	59	10	4	18	15	6921	1372	158	5862	7392
2009	102	5485	393	224	134	693	257	43	10	3	8	26	7378	1372	158	5965	7495
2010	110	5944	438	256	146	485	238	43	13	4	8	33	7718	1449	158	5962	7569
2011	114	6382	493	195	166	472	228	43	12	3	8	42	8169	1449	158	6304	7911
2012	116	6838	644	175	155	492	212	43	12	3	8	55	8762	1372	158	6322	7851
2013	127	7677	766	205	250	502	195	61	21	4	9	63	9886	1449	158	6733	8340
2014	126	7715	818	186	235	470	258	61	27	4	8	74	9990	1508	158	7195	8861
2015	121	7633	859	208	269	439	238	61	32	5	8	105	9985	1508	158	7472	9138
2016	131	7769	902	237	316	452	230	61	34	5	8	139	10285	1449	158	6675	8282
2017	131	7831	973	236	299	458	223	61	39	5	8	163	10429	1449	158	6788	8395
2018	130	8150	980	285	296	416	217	61	35	5	8	189	10774	1508	158	6550	8216
2019	125	8300	998	337	282	439	223	61	36	5	8	215	11028	1508	158	6705	8371

**Table 6 ijerph-19-16424-t006:** Statistics of the carbon emissions (tons of carbon/people, tons of carbon/million yuan).

Year	Carbon Emission Per Capita				Carbon Emission Intensity			
	Primary	Secondary	Tertiary	Per Capita	Primary	Secondary	Tertiary	Intensity
2000	0.039	5.011	0.179	0.727	0.107	2.759	0.114	1.416
2001	0.042	5.314	0.184	0.774	0.11	2.544	0.109	1.364
2002	0.041	5.032	0.192	0.778	0.103	2.39	0.105	1.264
2003	0.044	5.147	0.2	0.846	0.111	2.293	0.097	1.213
2004	0.046	4.824	0.217	0.828	0.09	1.954	0.099	1.024
2005	0.044	4.741	0.23	0.878	0.086	1.801	0.108	1.008
2006	0.046	5.057	0.244	0.968	0.085	1.72	0.108	0.982
2007	0.051	5.066	0.258	1.022	0.082	1.486	0.104	0.861
2008	0.061	5.177	0.268	1.128	0.085	1.38	0.104	0.837
2009	0.064	5.507	0.276	1.203	0.09	1.362	0.105	0.83
2010	0.069	5.848	0.302	1.296	0.083	1.125	0.101	0.705
2011	0.072	6.146	0.332	1.369	0.076	0.973	0.099	0.626
2012	0.076	6.175	0.411	1.463	0.076	0.975	0.12	0.63
2013	0.086	6.566	0.468	1.64	0.079	1.012	0.126	0.648
2014	0.089	6.371	0.486	1.642	0.076	0.97	0.122	0.613
2015	0.086	6.195	0.501	1.625	0.071	0.988	0.114	0.588
2016	0.095	6.238	0.521	1.66	0.075	0.959	0.104	0.556
2017	0.096	6.217	0.555	1.667	0.073	0.894	0.098	0.509
2018	0.096	6.452	0.554	1.704	0.072	0.847	0.083	0.464
2019	0.093	6.582	0.562	1.732	0.064	0.808	0.079	0.444

**Table 7 ijerph-19-16424-t007:** Unit root test for the residual series.

Item		Test-Statistic	Prob.
Augmented Dickey–Fuller test for unit root		−3.096	0.027
Test critical values	1% level	−3.75	None
	5% level	−3	None
	10% level	−2.63	None

**Table 8 ijerph-19-16424-t008:** Estimated change rates of various driving factors of social development.

Ranking	Period	U	S	T	A
slow	2020–2025	1.00%	1.60%	−5%	4.60%
	2026–2030	1.00%	1.50%	−4.80%	4.50%
	2031–2035	1.00%	1.40%	−4.60%	4.40%
	2036–2040	1.00%	1.30%	−4.40%	4.30%
medium	2020–2025	1.80%	2.60%	−6.50%	5.60%
	2026–2030	1.70%	2.50%	−6.40%	5.50%
	2031–2035	1.60%	2.40%	−6.30%	5.40%
	2036–2040	1.50%	2.30%	−6.20%	5.30%
fast	2020–2025	2.00%	3%	−7%	6%
	2026–2030	1.90%	2.90%	−6.90%	5.90%
	2031–2035	1.80%	2.80%	−6.80%	5.80%
	2036–2040	1.70%	2.70%	−6.70%	5.70%

## Data Availability

The data came from the literature and economic statistics and management practices, and all the data had literature sources indicated in the study.

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
