# Peer review of "The Carbon Emission Characteristics and Reduction Potential in Developing Areas: Case Study from Anhui Province, China"

_ijerph, 2022, doi:10.3390/ijerph192416424_

Round 1
Reviewer 1 Report
The subject picked by the authors is of great relevance in today's world. However they need to keep in mind certain points
a. Literature review must include studies from developed world where they can draw parallels to their study
b. Control variables specifically like R&D intensity, expenditure on renewable energy by state can bring more insights
c. Integrating Kuznet curve to the study is important as it presents as two parallel themes are part of the paper. How this is playing roles in terms of growth of carbon sink in Anhui province
d. Based on kuznet curve findings authors can specify the level of carbon emission and reduction corresponding to GDP growth of the Anhui province.
e. Instead of Ridge regression GMM can be used to get better results after correcting for the problem of collinearity.
The paper has lots of potential only it need to bring out clear relationship between carbon emission, sink and implication by Kuznet curve
Author Response
The reviewer’s issues and authors’ responses to Reviewer 1:
Comments and suggestions for authors: The subject picked by the authors is of great relevance in today's world. However they need to keep in mind certain points:
Issue a: Literature review must include studies from developed world where they can draw parallels to their study.
Response: Thanks for your kind comments and suggestions. We always keep in mind that literature review must include studies from developed world where we can draw parallels to the study. There are several studies from developed world in the revised manuscript, such as these cited references [1-14, 34, 38, 42, 44, 50, 51, 56, 57, 59, 61], and soon on and so forth.
Issue b: Control variables specifically like R&D intensity, expenditure on renewable energy by state can bring more insights.
Response: Thanks for your kind comments and suggestions. We agree that control variables specifically like R&D intensity, expenditure on renewable energy by state can usually bring more insights. There are specifically defined control variables on the social economic development (i.e. GDP, urbanization rate, and industrial structure) and the expenditure on renewable energy (like energy intensity and carbon intensity) by state, but no R&D intensity included due to the unavailability of complete data.
Issue c: Integrating Kuznet curve to the study is important as it presents as two parallel themes are part of the paper. How this is playing roles in terms of growth of carbon sink in Anhui province.
Response: Thanks for your kind comments and suggestions.
On the one hand, carbon sink refers to the main amount of carbon dioxide absorbed and stored by forests. In the study, carbon sources refer to the parent body that releases carbon into the atmosphere in nature, and carbon sink refers to the natural host body of carbon or the forests’ capacity to absorb and store carbon dioxide. The reduction of carbon sources is generally achieved through carbon dioxide emission reduction, while the technology of carbon sequestration is mainly used to increase the carbon sink sources.
Carbon sequestration is also called the carbon sink project, which refers to measures to increase the carbon content of carbon pools other than the atmosphere, including physical carbon sequestration and biological carbon sequestration. In December 1997, in order to mitigate the trend of global warming, representatives of 149 countries and regions adopted the so-called Kyoto Protocol in Kyoto, Japan, which officially entered into force on February 16, 2005. As a result, the international "carbon emission trading system" (referred to as "carbon sink project") has been formed. Particularly, the Kyoto Protocol, which aims to reduce global greenhouse gas emissions, is an international law of carbon sink project that limits carbon dioxide emissions of all countries in the world.
On the other hand, the special relationship between the state’s income and the deterioration of national environmental quality in a country will show an inverted U shaped curve, according to the hypothesis of Environmental Kuznets Curve (EKC). In other words, in the early stage of economic development, the environmental quality deteriorated slightly, but with the increase of national income, the environmental quality will gradually deteriorate; However, when the economic development reaches a turning point, the industrial structure gradually transforms from manufacturing to service or technology intensive industries, and the environmental quality will be improved. Though it has not been empirically proved that sustained economic development can automatically improve the deteriorating environmental quality, these regions with the most serious deterioration cases of environmental quality are usually developing countries with the rapid economic growth. Meanwhile, the developed countries have a better environmental quality.
In this study, although carbon sink and carbon sink project are not directly related to the environmental Kuznets curve, but they form an indirect logical link between the environmental governance and the technological innovation through economic development, carbon sources and carbon emissions in a developing country. Therefore, the growth of carbon sink and the implementation of carbon sink project may effectively promoted the improvement of the regional environmental governance and the development of technological innovation in Anhui Province, which will inevitably affect the direction of the environmental Kuznets curve.
Issue d: Based on Kuznet curve findings, authors can specify the level of carbon emission and reduction corresponding to GDP growth of the Anhui province.
Response: Thanks for your kind comments and suggestions. In the reported relationships between social economic development and regional environment quality, the environmental Kuznets curve is often used by researchers and scholars to judge whether there is a pattern of advanced pollution before treatment in regional economic development. In fact, in this study, there is an approximate inverted "U"-shaped Kuznets curve found between carbon emission and economic development in Anhui Province. Furthermore, the simultaneously discovered curve of the correlation between carbon emission and urbanization rate showed a positive "U" shape, which has economic implications too. The latter "positive U"-shaped correlation showed the great emission reduction potential with carbon sink under the environmental Kuznets curve in developing areas in Anhui Province, China. The reasons for these changes are properly that all cities and industrial districts in Anhui Province are promoting the new energy vehicles and industrial green technologies, whereas the provincial industrial and agricultural enterprises are following a national sustainable development pathway. Finally, based on the STIRPAT (stochastic impacts by regression on population, affluence, and technology) model, this study had expanded the relationship between the carbon emission intensity and the environmental Kuznets curve in Anhui Province, which further refined the change rules of the carbon emission intensity in Anhui Province, with more targeted suggestions.
Issue e: Instead of Ridge regression, GMM can be used to get better results after correcting for the problem of collinearity.
Response: Thanks for your kind comments and suggestions. In statistics, multicollinearity (or multiple collinearity) refers to that the explanatory variables of linear regression model are distorted/inaccurate or difficult to estimate accurately the correlation between variables due to precise correlation or high correlation. Multicollinearity will properly lead to functional failure of the specific prediction model, while the predicted interval of statistic model tends to become enlarged with the increase of variance to make prediction meaningless. In statistics, the common methods to eliminate multicollinearity are sorted in the following three types: (1) It is necessary to obtain additional new sample data to increase the sample size, if the multicollinearity issue is caused by the small number of samples; (2) The unimportant explanatory variables will be deleted from the regression model, if it is a multicollinearity problem caused by these variables; (3) If it is a multicollinearity issue caused by the model design, it is necessary to adopt some biased estimation models of regression coefficients to improve the regression model, such as the models of ridge regression, principal component, partial least squares, etc.
Ridge regression (or Tikhonov regulation) is a biased estimation regression method dedicated to solve the problem of collinear data, which is essentially an improved least squares estimation method. Ridge regression is known as the most frequently used regularization method for regression analysis of ill posed problems and multicollinearity issue in the regression model. By giving up the unbiasedness of the least squares method, ridge regression obtains regression coefficients at the cost of losing part of the information and reducing the accuracy, which is more practical and reliable, and the fitting of missing data is better than the least squares method.
The endogeneity indicates one or more explanatory variables are related to the random disturbance term in a given panel data regression model. The explanatory variable and the explained variable interact, influence and cause each other in the model. Thus, no matter how large the sample size is, the estimator of the regression model will not converge to the true parameter value. In econometrics, all the explanatory variables related to the disturbance error terms may be called "endogenous variables" due to the following endogenous sources in theory, i.e. (1) the issue of missing variable deviation, (2) the classical measurement error problem, and (3) the simultaneous reverse causality.
Firstly, this study employs the method of SYS-GMM (System Generalized Method of Moment) estimator to test and delete the possible endogenous effects among variables. The SYS-GMM estimator method solves some missing errors in differential GMM. By adding the lag term of the explained variables, the study establish the specific measurement model based on the SYS-GMM estimation method. In fact, the SYS-GMM estimator method is applicable here regardless of whether there are endogenous explanatory variables in the regression model. The results of the new regression model showed that the lag coefficient of lnI was 0.366, which was significant at the level of 1%. The estimated P values corresponding to the lag term AR (1) and AR (2) are 0.085 and 0.553, respectively. Meanwhile, the P value of Sargan test was 0.685, which indicated that the econometric model had a dynamic relationship and can be estimated using the system GMM method. The lagging term of carbon emission intensity was estimated as 0.366 at the significant level of 1%, indicating that carbon emission intensity was positively affected by the previous study period. The carbon emission intensity of the previous period has an inertial effect on the investigated years. Results showed that the primary variables except energy intensity passed the test at the significant level of 1%. The coefficients of per capita GDP and its square term were 1.476 and -0.12, respectively. Thus, it showed an inverted "U" curve too, which indicated that there was an environmental Kuznets curve indeed for the carbon emission intensity in Anhui Province. The first term coefficient of per capita GDP was positive, indicating that the current economic development has promoted the carbon emission intensity in Anhui Province.
Furthermore, in the study, some economic data (such as the urbanization rate) cannot meet specific requirements of the SYS-GMM estimator method, due to the high data requirements of GMM methods. Actually, we have tried many times both the methods of GMM and ridge regression and the other regression models. We could only use the SYS-GMM estimator method in analyzing the environmental Kuznets curve for carbon emissions in Anhui Province. Therefore, we combined the methods of the ridge regression model and the SYS-GMM estimator to estimate the carbon emission potential in Anhui Province.
Finally, we believed that it was best and more appropriate to adopt both the ridge regression model and the SYS-GMM estimator method in analyzing the environmental Kuznets curve for carbon emissions with those results obtained in Anhui Province.
Issue f: The paper has lots of potential only it need to bring out clear relationship between carbon emission, sink and implication by Kuznet curve
Response: Thanks for your kind comments and suggestions. The relationships among carbon emission and carbon sink and the implication by Kuznet curve are now clearly presented through both the ridge regression model and the SYS-GMM estimator methods, as well as the revised words in literature review. Please see the above authors’ responses to issues c and d.

Reviewer 2 Report
The article addresses a very interesting topic. The article is well structured, the ideas are presented in a logical, concise order. The statements are supported by statistical data, that are analysed in a proper way by the authors.
The article has the potential to be published, which is why I recommend some revisions
1. In introduction, authors can mentions some of the achievements of the European Union in energy transition and carbon neutrality process.
https://www.europarl.europa.eu/news/en/headlines/society/20180305STO99003/reducing-carbon-emissions-eu-targets-and-measures
https://ec.europa.eu/eurostat/web/products-eurostat-news/-/ddn-20210507-1
https://www.europarl.europa.eu/news/en/headlines/society/20190926STO62270/what-is-carbon-neutrality-and-how-can-it-be-achieved-by-2050
2. The authors made a very good analysis of the situation in the Anhui region, but they should better motivate the choice of this area.
3. The authors must better justify the choice of the method used.
4. In the conclusions section, the authors must present the limits of research and future directions for their research
Author Response
The reviewer’s issues and authors’ responses to Reviewer 2:
Comments and suggestions for authors: The article addresses a very interesting topic. The article is well structured, the ideas are presented in a logical, concise order. The statements are supported by statistical data that are analysed in a proper way by the authors. The article has the potential to be published, which is why I recommend some revisions
Issue 1: In introduction, authors can mentions some of the achievements of the European Union in energy transition and carbon neutrality process.
https://www.europarl.europa.eu/news/en/headlines/society/20180305STO99003/reducing-carbon-emissions-eu-targets-and-measures
https://ec.europa.eu/eurostat/web/products-eurostat-news/-/ddn-20210507-1
https://www.europarl.europa.eu/news/en/headlines/society/20190926STO62270/what-is-carbon-neutrality-and-how-can-it-be-achieved-by-2050
Response: Thanks for your kind comments and suggestions. The achievements of the European Union in energy transition and carbon neutrality process are introduced and explored in introduction with the suggested references. Additionally, we also added and mentioned the following informational website entries.
Each Country's Share of CO2 Emissions. Available online: https://www.ucsusa.org/resources/each-countrys-share-co2-emissions (accessed on 14-01-2022).
EU economy greenhouse gas emissions -24% since 2008. Available online: https://ec.europa.eu/eurostat/en/web/products-eurostat-news/-/ddn-20221115-2 (accessed on 10‑01‑2022).
CO2 emissions from energy use clearly decreased in the EU in 2020. Available online: https://ec.europa.eu/eurostat/web/products-eurostat-news/-/ddn-20210507-1 (accessed on 07‑05‑2021).
United Kingdom: CO2 Country Profile. Available online: https://ourworldindata.org/co2/country/united-kingdom?country=~GBR (accessed on 01‑08‑2020).
United States: CO2 Country Profile. Available online: https://ourworldindata.org/co2/country/united-states?country=~USA (accessed on 01‑08‑2020).
Issue 2: The authors made a very good analysis of the situation in the Anhui region, but they should better motivate the choice of this area.
Response: Thanks for your kind comments and suggestions. The choice of this area is much more motivated now.
Issue 3: The authors must better justify the choice of the method used.
Response: Thanks for your kind comments and suggestions. The choice of the method used is much more justified.
Issue 4: In the conclusions section, the authors must present the limits of research and future directions for their research.
Response: Thanks for your kind comments and suggestions. We added the limits of research and future directions for the forward research. Please see the last part “Remarks on the limitations and future directions for the research”.

Reviewer 3 Report
This paper focuses on the carbon emission characteristics and reduction potential of
Anhui province in China. Estimation methods of each kind of carbon emissions are
clearly given. Then, based on the estimated values, some econometric models are used
to explore the relation between resident income and the environment. Obviously, the
authors have made great efforts in study, but there are too many minor issues in this
paper.
Major issues:
(1) Language should be improved. There are too many minor issues and I have listed
some of the minor issues. Please carefully check and modify them.
(2) In section 2.2.3, the first paragraph, except agricultural machinery, there are some
other sources of carbon emissions, such as the production and use of the chemical
fertilizer. And, as I known, this is much more the carbon emission caused by the use of
agricultural machinery.
(3) Tables especially the Table 2, 3, 5, 7 need to be rearranged since the contents of
some of they are out of the page range, and some of them make me uncomfortable (such
as Table 7).
(5) Compared with equation (13) and (14), why you add two new terms in equation (15)
and (16)?
(6) Ridge regression is not designed for collinearity. It is designed for variable selection.
For the issue of variable selection, most recent studies prefer to use the LASSO model.
(7) Environmental Kuznets Curve is used to describe the relation between income per
capita (or GDP per capita) and the quality of the environment. Rather than the aggregate
income and the environment. Am I right? If I am correct, please modify the related
contents.
(8) The number of observations in your regression analysis is 20 (from 2000 to 2020),
am I right? If I am correct, there is a very fatal problem, that is, n=20 is too small (most
of the statistical and econometric textbooks suggest that the n should be greater than 30
or 50). The limited n means that estimators are not asymptotic and converged even we
assume that all the effects are time invariant.
(9) Figures especially Figure 1 need to be improved. For example, they are not quite
clear; the values in are grey rather than black. Particularly, if the space is enough for
the tables or figures, please centered them on the content rather than the page.
(10) The influence of wind is neglected in this study. It should have effect on the carbon
issue.
Minor issues:
(1) In section 2.2.1, the first paragraph, “ith” or “i-th”? Informal language. Please
modify all the same abbreviations.
(2) In section 2.2.1, the second paragraph, “u” or “μ”?
(3) In section 2.2.1, the third paragraph, <1> since there is no Abbreviation section at
the end, all the unusual scales such as “kgce”, “tco2” should be described or footnoted.
<2> some of your “i” are italic but some are not. Unify their forms in all the following
contents.
(4) In section 2.2.2, the second paragraph, “be showed” is incorrect expression, and
should be “be shown”.
(5) Although there are no mandatory requirements, the formulas are usually called as
“equation 1, 2, 3, ...” rather than “formula 1, 2, 3, ...”.
(6) In Table 1, some nouns are singular and some are plural.
(7) For equation (6) as well as the following paragraph, why C2 is uppercase but c1 is
lowercase?
(8) Equation (12) is twisted.
(9) In section 2.4.1, the second paragraph, some letters are italic but some are not? Some
as the issues in equation (13) and (14), and the “U” in (15).
(10) In section 2.4.2, the first paragraph, why there is a jump of the word “prevent”?
And a jump of the word “co-integration” in the second paragraph of section 2.4.2?
[There also have many minor issues that I found after section 2.4.2 but I really do not
want list them one by one.]
Author Response
The reviewer’s issues and authors’ responses to Reviewer 3:
Comments and suggestions for authors: This paper focuses on the carbon emission characteristics and reduction potential of Anhui province in China. Estimation methods of each kind of carbon emissions are clearly given. Then, based on the estimated values, some econometric models are used to explore the relation between resident income and the environment. Obviously, the authors have made great efforts in study, but there are too many minor issues in this paper.
Major issues:
Issue (1): Language should be improved. There are too many minor issues and I have listed some of the minor issues. Please carefully check and modify them.
Response: Thanks for your kind comments and suggestions. We have carefully checked the manuscript and modify them.
Issue (2): In section 2.2.3, the first paragraph, except agricultural machinery, there are some other sources of carbon emissions, such as the production and use of the chemical fertilizer. And, as I known, this is much more the carbon emission caused by the use of agricultural machinery.
Response: Thanks for your kind comments and suggestions. In the study, carbon sources refer to the parent body that releases carbon into the atmosphere in nature, including physical, chemical, and biological carbon sources. Therefore, carbon sources are not only the carbon emission caused by the use of agricultural machinery. The reduction of carbon sources is generally achieved through carbon dioxide emission reduction, including reduced usage of carbon sources and carbon sequestration from these various carbon sources.
Issue (3): Tables especially the Table 2, 3, 5, 7 need to be rearranged since the contents of some of they are out of the page range.
Response: Thanks for your kind comments and suggestions. These tables are rearranged now.
Issue (4): ……, and some of them make me uncomfortable (such as Table 7).
Response: Thanks for your kind comments and suggestions. These tables are rearranged now.
Issue (5): Compared with equation (13) and (14), why you add two new terms in equation (15) and (16)?
Response: Thanks for your kind comments and suggestions. In unit root test, the ADF (Augmented Dickey Fuller) method can be used to test whether there is a higher-order lag correlation in time series data. In the ADF formulas, i.e. equations (15) and (16), the first order difference (or even the second order difference sometimes) is required before analyzing the original data to check whether whether there is a higher-order lag correlation in the studied data. Therefore, there are additionally two new terms in equations (15) and (16) due to the first order difference.
Issue (6): Ridge regression is not designed for collinearity. It is designed for variable selection. For the issue of variable selection, most recent studies prefer to use the LASSO model.
Response: Thanks for your kind comments and suggestions. In traditional statistics, multicollinearity (or multiple collinearity) refers to that the explanatory variables of linear regression model are distorted/inaccurate or difficult to estimate accurately the correlation between variables due to precise correlation or high correlation. Multicollinearity will properly lead to functional failure of the specific prediction model, while the predicted interval of statistic model tends to become enlarged with the increase of variance to make prediction meaningless. In statistics, the common methods to eliminate multicollinearity are sorted in the following three types: (1) To obtain additional new sample data to increase the sample sizes; (2) To delete unimportant explanatory variables in the regression models; (3) To adopt some biased estimation models of regression coefficients to improve the regression model, such as the models of ridge regression, principal component, partial least squares, etc. Ridge regression (or Tikhonov regulation) is a biased estimation regression method dedicated to solve the problem of collinear data, which is essentially an improved least squares estimation method. Ridge regression is known as the most frequently used regularization method for regression analysis of ill posed problems and multicollinearity issue in the regression model. By giving up the unbiasedness of the least squares method, ridge regression obtains regression coefficients at the cost of losing part of the information and reducing the accuracy, which is more practical and reliable, and the fitting of missing data is better than the least squares method.
In essence, ridge regression is an improved least squares estimation method. By giving up the unbiased nature of the least squares method, it is more practical and reliable to obtain regression coefficients at the cost of losing part of the information and reducing the model’s precision. However, the fitting of ridge regression for ill conditioned data is stronger than the traditional least squares method. The Lasso (Least Absolute Selection and Shrinkage Operator) regression is actually the minimum absolute value selection and contraction operator. Lasso regression is a good compression estimation method based on the idea of reducing variable set (reducing order). By constructing a penalty function, it can compress the coefficients of variables and make some regression coefficients become zero, thus achieving the purpose of variable selection. Both the two regression methods can be used to solve the over fitting problem of standard linear regressions. Since Lasso regression solves the problem of over fitting through variable selection, but the selected variables related to carbon emission intensity in this study are all these core variables obtained by referring to previously reports and published papers, which have a strong impact on carbon emission intensity and this impact cannot be deleted. Therefore, we finally adopted the ridge regression model with those results reported in this study.
Issue (7): Environmental Kuznets Curve is used to describe the relation between income per capita (or GDP per capita) and the quality of the environment. Rather than the aggregate income and the environment. Am I right? If I am correct, please modify the related contents.
Response: Thanks for your kind comments and suggestions. Yes, the environmental Kuznets curve is used to describe this relation between income per capita (or GDP per capita) and the quality of the environment. Nevertheless, the study mainly explored the relation between income per capita (or GDP per capita) and energy consumption and the quality of the environment, instead of that relation between the aggregate income and the environment. In the reported relationships between social economic development and regional environment quality, the environmental Kuznets curve is often used by researchers and scholars to judge whether there is a pattern of advanced pollution before treatment in regional economic development. In fact, in this study, there is an approximate inverted "U"-shaped Kuznets curve found between carbon emission and economic development in Anhui Province. Furthermore, the simultaneously discovered curve of the correlation between carbon emission and urbanization rate showed a positive "U" shape, which has economic implications too.
Issue (8): The number of observations in your regression analysis is 20 (from 2000 to 2020), am I right? If I am correct, there is a very fatal problem, that is, n=20 is too small (most of the statistical and econometric textbooks suggest that the n should be greater than 30 or 50). The limited n means that estimators are not asymptotic and converged even we assume that all the effects are time invariant.
Response: Thanks for your kind comments and suggestions. It should be noted that the statistics and econometrics textbooks suggest that the sample size of a large sample should be no less than 30 (n≥30), while that of the small sample is no more than 30 (n≮30). The number of observations (samples observed) in this study’s regression analysis is 20*5=100 (20 years multiplied by 5 duplicates of sample sources), i.e. n=100. The 5 duplicates of sample sources are derived from these five different types of carbon emission sources sampled from 2000 to 2020, i.e. the sources of carbon emissions from energy consumption, carbon emissions from food consumption of the residents, carbon emissions from cultivated land, carbon emissions from ruminants, and carbon emissions from waste (including waste materials and wastewater). Therefore, the total sample size (n) of this study is not 20 but 100 (n=100). Actually, there are also some statistical approaches applicable to analyze small samples (n≮30), such as Fisher's precision probability test (or Fisher's exact test) and the nonparametric estimation methods. In this study, the samples observed are 100, and there is no statistic problem of the limited sample size.
Issue (9): Figures, especially Figure 1 need to be improved. For example, they are not quite clear; the values in are grey rather than black. Particularly, if the space is enough for the tables or figures, please centered them on the content rather than the page.
Response: Thanks for your kind comments and suggestions. It is amended and improved now.
Issue (10): The influence of wind is neglected in this study. It should have effect on the carbon issue.
Response: Thanks for your kind comments and suggestions. Yes, the influence of wind is indeed existing in the research. However, its data or datasets are difficult to be obtained and acquired for a fine study.
Minor issues:
Issue (1): In section 2.2.1, the first paragraph, “ith” or “i-th”? Informal language. Please modify all the same abbreviations.
Response: Thanks. It’s revised.
Issue (2): In section 2.2.1, the second paragraph, “u” or “μ”?
Response: Thanks. It’s revised.
Issue (3): In section 2.2.1, the third paragraph, <1> since there is no Abbreviation Section at the end, all the unusual scales such as “kgce”, “tco2” should be described or footnoted. <2> some of your “i” are italic but some are not. Unify their forms in all the following contents.
Response: Thanks. They are properly revised and amended.
Issue (4): In section 2.2.2, the second paragraph, “be showed” is incorrect expression, and should be “be shown”.
Response: Thanks. It’s revised.
Issue (5): Although there are no mandatory requirements, the formulas are usually called as “equation 1, 2, 3, ...” rather than “formula 1, 2, 3, ...”.
Response: Thanks. They are properly revised and amended.
Issue (6): In Table 1, some nouns are singular and some are plural.
Response: Thanks. They are properly revised and amended.
Issue (7): For equation (6) as well as the following paragraph, why C2 is uppercase but c1 is lowercase?
Response: Thanks. It’s revised.
Issue (8): Equation (12) is twisted.
Response: Thanks. It’s revised and amended.
Issue (9): In section 2.4.1, the second paragraph, some letters are italic but some are not? Some as the issues in equation (13) and (14), and the “U” in (15).
Response: Thanks. These italic letters in equations (13-15) were actually written by the WORD formula editor due to the auto generated status, which is now re-written and revised now.
Issue (10): In section 2.4.2, the first paragraph, why there is a jump of the word “prevent”? And a jump of the word “co-integration” in the second paragraph of section 2.4.2?
Response: Thanks. The word “prevent” is revised as “avoid”, while the word “co-integration” needs no revision since the EG two-step method was used to explore whether there was a co-integration relationship between different variables in unit root test and co-integration test.
Issue (11): There also have many minor issues that I found after section 2.4.2 but I really do not want list them one by one.
Response: Thanks. They are carefully checked and revised now.

Reviewer 4 Report
The authors have examined different characteristics of carbon emissions by selecting the sample of Anhui province, China. The Paper is quite interesting, however, there are several deficiencies in it, which can be removed.
1. Abstract is very poorly written. I suggest improving the abstract by following this sequence: most attractive introductory statement, the purpose of the study, sample period, methods, results, and suggestions.
2. In the introduction, the authors have just mentioned the EKC theory, which is not enough. There are several theories that can support this study. Theoretically, this paper is so weak. So, I suggest adding a separate section "theoretical framework and literature review" in it.
3. In the methodological section, the authors have mentioned that they have collected data for the time period 2001-2020, but they have mentioned different study time period in other sections.
4. In the methodology, authors have used unit root and co-integaration tests and ridge regression. I suggest adding a robustness test to find more robust results.
5. The authors have used panel data, which has the potential for endogeneity issue. I suggest addressing this issue in this paper with an appropriate method.
6. Limitations of the study and recommendations for future research are missing in the last section. I suggest adding both of these.
Author Response
The reviewer’s issues and authors’ responses to Reviewer 4:
Comments and suggestions for authors: The authors have examined different characteristics of carbon emissions by selecting the sample of Anhui province, China. The Paper is quite interesting, however, there are several deficiencies in it, which can be removed.
Issue 1: Abstract is very poorly written. I suggest improving the abstract by following this sequence: most attractive introductory statement, the purpose of the study, sample period, methods, results, and suggestions.
Response: Thanks for your kind comments and suggestions. The abstract is properly revised and amended in the revised version.
Issue 2: In the introduction, the authors have just mentioned the EKC theory, which is not enough. There are several theories that can support this study. Theoretically, this paper is so weak. So, I suggest adding a separate section "theoretical framework and literature review" in it.
Response: Thanks for your kind comments and suggestions. We cannot add an additional separate section for the following reasons: (1) We could not change the journal’s publication style requested by the given template of International Journal of Environmental Research in Public Health (IJERPH). According to periodical regulations of IJERPH, the typical parts of a research article are mainly as follows: 1. Introduction, 2. Materials and Methods, 3. Results, 4. Discussion, and 5. Conclusions. (2) There is a sufficient literature review in this paper, and we do not want to repeat the redundant work of literature review again. (3) Alternatively, we choose to expand the development of EKC theory in the part “materials and methods” to strengthen the paper’s theoretical content.
Issue 3: In the methodological section, the authors have mentioned that they have collected data for the time period 2001-2020, but they have mentioned different study time period in other sections.
Response: Thanks for your kind comments and suggestions. In the materials and methods part, we mentioned that the official version of the statistical yearbook were collected from 2001 to 2020. We did not state that the statistical data were collected for analysis from 2001 to 2020. In the other parts, we claimed the research period during 2000 to 2019 and the statistical data from 2000 to 2019. There is no contradiction between the two developments. This is because the annual data recorded in the statistical yearbook officially released in each year is one year behind. For example, the 2001 statistical yearbook records the data of 2000, and so on, whereas the 2020 statistical yearbook records the data of 2019. The statistical data adopted and analyzed are retrieved from the official version of the statistical yearbook from 2001 to 2020, which is actually the statistical data from 2000 to 2019 used in this study.
Issue 4: In the methodology, authors have used unit root and co-integaration tests and ridge regression. I suggest adding a robustness test to find more robust results.
Response: Thanks for your kind comments and suggestions. In this study, unit root and co-integration analysis tests are used to avoid the collinearity of variables in the regression model and make the regression model used more stable. At the same time, we used another conversion method to replace the core variables for robustness test, and regressed the indicators of carbon emissions as the explained variables. Herein, the later results obtained are the same as the original conclusions, which proves that the regression model has a much good robustness. The results of the latter conversion method of robustness test are consistent with the results of unit root and co-integration analysis. This latter content of robustness test is not included in the revised paper to avoid redundancy.
Issue 5: The authors have used panel data, which has the potential for endogeneity issue. I suggest addressing this issue in this paper with an appropriate method.
Response: Thanks for your kind comments and suggestions. This issue has been solved by hard work. Generally, the error term of panel data regression model is composed of two parts. One is related to individual observation unit. It summarizes all the factors that affect the explained variables, but do not change with time. The other summarizes the unobservable factors that change with time due to the section, which are usually called the specific error or the specific disturbance term error (in fact, the second part of the error can also be divided into two sub-parts again). If the panel data regression model is endogenous, the explanatory variable related to the error item is defined as "endogenous variable". On the contrary, if it’s unrelated to the error term, it is named "exogenous variable". The serious consequence of endogenous variables is that the estimator of the regression model will be inconsistent. In the case, there is the so-called endogeneity in the regression model with endogenous variables. The endogeneity indicates one or more explanatory variables are related to the random disturbance term in a given panel data regression model. The explanatory variable and the explained variable interact, influence and cause each other in the model. Thus, no matter how large the sample size is, the estimator of the regression model will not converge to the true parameter value. In econometrics, all the explanatory variables related to the disturbance error terms may be called "endogenous variables" due to the following endogenous sources in theory, i.e. (1) the issue of missing variable deviation, (2) the classical measurement error problem, and (3) the simultaneous reverse causality. This study employs the method of SYS-GMM (System Generalized Method of Moment) estimator to test and delete the possible endogenous effects among variables. The SYS-GMM estimator method solves some missing errors in differential GMM. By adding the lag term of the explained variables, the study establish the specific measurement model based on the SYS-GMM estimation method. In fact, the SYS-GMM estimator method is applicable here regardless of whether there are endogenous explanatory variables in the regression model.
On the basis of the analyses with the STIRPAT regression model, the square term of GDP per capita variable is added to test whether there is an environmental Kuznets curve revisited later on carbon emissions in Anhui Province. In order to exclude the possible emerging endogeneity of panel data regression model to ensure the predicting accuracy of the regression model, the study initially conducted ADF test on the square term of per capita GDP variable. It was found that there was a non-stationary series and the corresponding first-order difference is a stationary series. At the same time, this study used the EG two-step method to carry out cointegration tests on the equation, and found that it passes the linear cointegration test at 10% level. In order to exclude the possible endogenous effects among later added variables, the SYS-GMM estimator method was further adopted to detect all the possible endogenous variables. Then, a new regression model was established by adding the lag term of the explained variables. Subsequently, the study analyzed and explored it based on the SYS-GMM estimator method. The results of the new regression model showed that the lag coefficient of lnI was 0.366, which was significant at the level of 1%. The estimated P values corresponding to the lag term AR (1) and AR (2) are 0.085 and 0.553, respectively. Meanwhile, the P value of Sargan test was 0.685, which indicated that the econometric model had a dynamic relationship and can be estimated using the system GMM method. The lagging term of carbon emission intensity was estimated as 0.366 at the significant level of 1%, indicating that carbon emission intensity was positively affected by the previous study period. The carbon emission intensity of the previous period has an inertial effect on the investigated years.
Results showed that the primary variables except energy intensity passed the test at the significant level of 1%. The coefficients of per capita GDP and its square term were 1.476 and -0.12, respectively. Thus, it showed an inverted "U" curve too, which indicated that there was an environmental Kuznets curve indeed for the carbon emission intensity in Anhui Province. Furthermore, the first term coefficient of per capita GDP was positive, indicating that the current economic development has promoted the carbon emission intensity in Anhui Province.
Finally, we believed that it was best and more appropriate to adopt both the ridge regression model and the SYS-GMM estimator method in analyzing the environmental Kuznets curve for carbon emissions with those results obtained in Anhui Province. Therefore, we combined the methods of ridge regression model and the SYS-GMM estimator in the study.
Issue 6: Limitations of the study and recommendations for future research are missing in the last section. I suggest adding both of these.
Response: Thanks for your kind comments and suggestions. We added the limits of the study and recommendations for the future research. Please see the last part “Remarks on the limitations and future directions for the research”.
